https://doi.org/10.1038/s41467-021-26233-8　　**OPEN**

# Coupled protein synthesis and ribosome-guided piRNA processing on mRNAs

Yu H. Sun [1], Ruoqiao Huiyi Wang[1], Khai Du[1], Jiang Zhu [1], Jihong Zheng[2], Li Huitong Xie[1], Amanda A. Pereira[1], Chao Zhang [2], Emiliano P. Ricci[3] & Xin Zhiguo Li [1✉]

PIWI-interacting small RNAs (piRNAs) protect the germline genome and are essential for fertility. piRNAs originate from transposable element (TE) RNAs, long non-coding RNAs, or 3′ untranslated regions (3′UTRs) of protein-coding messenger genes, with the last being the least characterized of the three piRNA classes. Here, we demonstrate that the precursors of 3′UTR piRNAs are full-length mRNAs and that post-termination 80S ribosomes guide piRNA production on 3′UTRs in mice and chickens. At the pachytene stage, when other co-translational RNA surveillance pathways are sequestered, piRNA biogenesis degrades mRNAs right after pioneer rounds of translation and fine-tunes protein production from mRNAs. Although 3′UTR piRNA precursor mRNAs code for distinct proteins in mice and chickens, they all harbor embedded TEs and produce piRNAs that cleave TEs. Altogether, we discover a function of the piRNA pathway in fine-tuning protein production and reveal a conserved piRNA biogenesis mechanism that recognizes translating RNAs in amniotes.

[1] Center for RNA Biology: From Genome to Therapeutics, Department of Biochemistry and Biophysics, University of Rochester Medical Center, Rochester, NY 14642, USA. [2] School of Life Sciences and Technology, Tongji University, Shanghai 200092, China. [3] Université de Lyon, ENSL, UCBL, INSERM, CNRS, LBMC, 46 Allée d'Italie, Lyon 69007, France. ✉email: Xin_Li@urmc.rochester.edu

PIWI-interacting RNAs (piRNAs) are a class of small silencing RNAs that are essential for animal fertility. piRNAs guide a specialized class of Argonaute proteins primarily found in germ cells, known as PIWI proteins[1–6], to target RNAs via base-pair complementarity. Unlike microRNAs (miRNAs) and small interfering RNAs (siRNAs) that are derived from precursors with double-stranded RNA structure, most piRNAs are derived from long single-stranded transcripts[7,8]. The 5′-monophosphate ends (5′P) of piRNAs are formed when piRNA precursors are fragmented into pre-piRNAs that are loaded onto PIWI proteins. This fragmentation process requires MOV10L1, a UPF1-like RNA helicase, and PLD6, an RNA endonuclease on mitochondrial outer membranes[9–16]. The PIWI-loaded pre-piRNAs are further trimmed by a 3′-to-5′ exonuclease (PNLDC1 in mice), resulting in a length distribution characteristic of each particular PIWI protein[17–26]. Finally, HEN1 binds PIWI proteins and adds a 2′-O-methyl group to the 3′-end of mature piRNA[27,28]. The resulting primary piRNAs can be further amplified via the ping-pong pathway[7,29], yielding secondary piRNAs.

Compared to our understanding of the piRNA pathway in fruit flies, it remains unclear what marks a transcript for piRNA processing in mammals. In fruit flies, piRNA loci in germ cells are epigenetically marked by heterochromatin-bound factor Rhino[30–32]. piRNA precursors are derived from promoter-independent, unspliced cryptic transcripts in highly repetitive regions that harbor diverse transposable elements (TEs)[33,34]. piRNA precursors are then directly channeled for piRNA processing on perinuclear RNA granules that are proximal to nuclear pores[35]. Maternally deposited piRNAs further recognize the piRNA precursors post-transcriptionally to trigger a cascade of piRNA processing[36,37]. Unlike germ cells in fruit flies predetermined by maternal deposition, mouse germ cells are induced from somatic cells[38,39], leaving parental piRNA deposition unlikely. The majority of mammalian piRNAs come from 5′ capped and 3′ polyadenylated, long continuous non-coding RNAs (lncRNAs) that are depleted of TEs[8]. The transcription factor A-MYB directs the expression of these lncRNAs from euchromatin regions without identifiable epigenetic markers distinguishing mouse piRNA loci, along with mRNAs that do not produce piRNAs, during pachynema (the pachytene stage of meiosis)[8,40]. Despite these piRNA precursors being annotated as lncRNAs, rather than directly channeling to RNA granules for processing, we recently demonstrated that ribosomes translate their upstream open reading frames (uORFs). Thus, transcription and translation of mammalian piRNA precursors utilize conventional machineries, therefore leaving the features that distinguish them from other lncRNAs and mRNAs elusive.

In eukaryotes, translation-dependent RNA quality-control mechanisms identify and degrade aberrant transcripts with premature termination codons (nonsense-mediated decay (NMD)), lacking in-frame stop codons (non-stop decay), or containing elongation inhibitory structures (no-go decay). These quality-control pathways are coupled with translation and facilitate the recycling of ribosomes trapped on faulty transcripts[41–44]. The finding of ribosome-guided piRNA biogenesis downstream of uORFs on lncRNAs[45] introduces the possibility that aberrant translation intermediates could act as substrates for piRNA biogenesis. It nevertheless remains unclear how 80S ribosomes assemble on the non-coding regions of piRNA precursors, which contain frequent stop codons, and how these ribosomes escape the translation-dependent surveillance mechanisms that recycle ribosomes and degrade RNAs[44]. The detection of ribosomes downstream of uORFs on lncRNA piRNA precursors is reminiscent of the re-initiation of ribosomes downstream of uORFs present in the 5′ untranslated regions (5′UTRs) of canonical

mRNAs when initiation factors have not yet dissociated from the 40S subunit[46]. Given that short uORF length is an important factor for translation of the main ORFs on mRNAs[46], which explains why eukaryotic mRNAs are generally monocistronic, the presence of long ORF on piRNA precursors should inhibit ribosome-guided piRNA biogenesis. It is thus mechanistically essential for both piRNA biogenesis and translational regulation to test whether ribosome-guided biogenesis can only occur after the translation of a short ORF.

To test whether a short uORF is required for ribosome-guided piRNA biogenesis, we studied a specific class of piRNAs, namely, 3′UTR piRNAs. In mice, piRNAs are divided into three major classes based on their origin[47,48]: (i) piRNAs from TEs (TE piRNAs), (ii) intergenic piRNAs derived from lncRNAs (pachytene piRNAs), and (iii) genic piRNAs that map to 3′UTRs of protein-coding regions in the sense orientation (3′UTR piRNAs). TE piRNAs protect the animal germline genome from TEs and are essential for animal fertility, a function that is evolutionarily conserved in bilateral animals[1–3,5,49,50]. Pachytene piRNAs have only been reported in mammals thus far and have been shown to trigger the decay of mRNAs required for sperm formation[51–54]. 3′UTR piRNAs have been detected in fruit flies, frogs, and diverse mammalian species[55–57], but we currently do not know their function(s), how their production is regulated, nor whether their precursors are full-length mRNAs or cryptic transcripts corresponding exclusively to the 3′UTR[56]. The lack of understanding of 3′UTR piRNAs hindered our efforts to identify either common features that mark such transcripts for piRNA biogenesis and/or machinery that sorts diverse RNAs for piRNA biogenesis.

Here we characterize the biogenesis of 3′UTR piRNAs in mice, demonstrating that their precursors are full-length protein-coding mRNAs. We further show that piRNA biogenesis from these precursors is coupled with efficient translation and that ribosomes guide piRNA precursor fragmentation on mRNA 3′UTRs. We demonstrate that this tight coupling of ribosome binding and piRNA biogenesis fine-tunes protein synthesis from mRNAs. Ribosome-guided piRNA processing occurs at the meiotic stage when ribosome recycling factors and NMD are temporally inhibited. Lastly, we demonstrate that 3′UTR ribosome-guided piRNA processing also occurs in chickens. Although 3′UTR piRNAs are derived from distinct sets of genes in mice and chickens, we found the presence of TE sequences to be a shared feature that serves to produce anti-sense TE piRNAs that cleave TEs post-transcriptionally, indicating that TE suppression is a conserved evolutionary force driving 3′UTR piRNA production. Taken together with our previous studies, we find that a general and conserved piRNA biogenesis pathway recognizes translating RNAs regardless of their ORF length.

## Results

**3′UTR piRNAs are produced from full-length mRNAs.** To test whether 3′UTR piRNAs are derived from full-length mRNAs or cryptic transcripts with alternative transcription start sites (TSSs), we set out to identify the structure of the precursor RNAs by blocking their transcription and inhibiting the post-transcriptional processing of 3′UTR piRNAs. We previously defined a group of 3′UTR piRNAs increasing from 12.5 days postpartum (dpp) to 17.5 dpp in mice[8], when spermatocytes enter pachynema (Supplementary Fig. 1a). Thus, we hypothesized that these 3′UTR piRNA precursors may be controlled by the transcription factor A-MYB, which also promotes the synthesis of pachytene piRNA precursors[8]. Chromatin immunoprecipitation sequencing (ChIP-seq) revealed that A-MYB-binding sites are far from the 3′UTRs of mRNAs that come from 3′UTR piRNA-

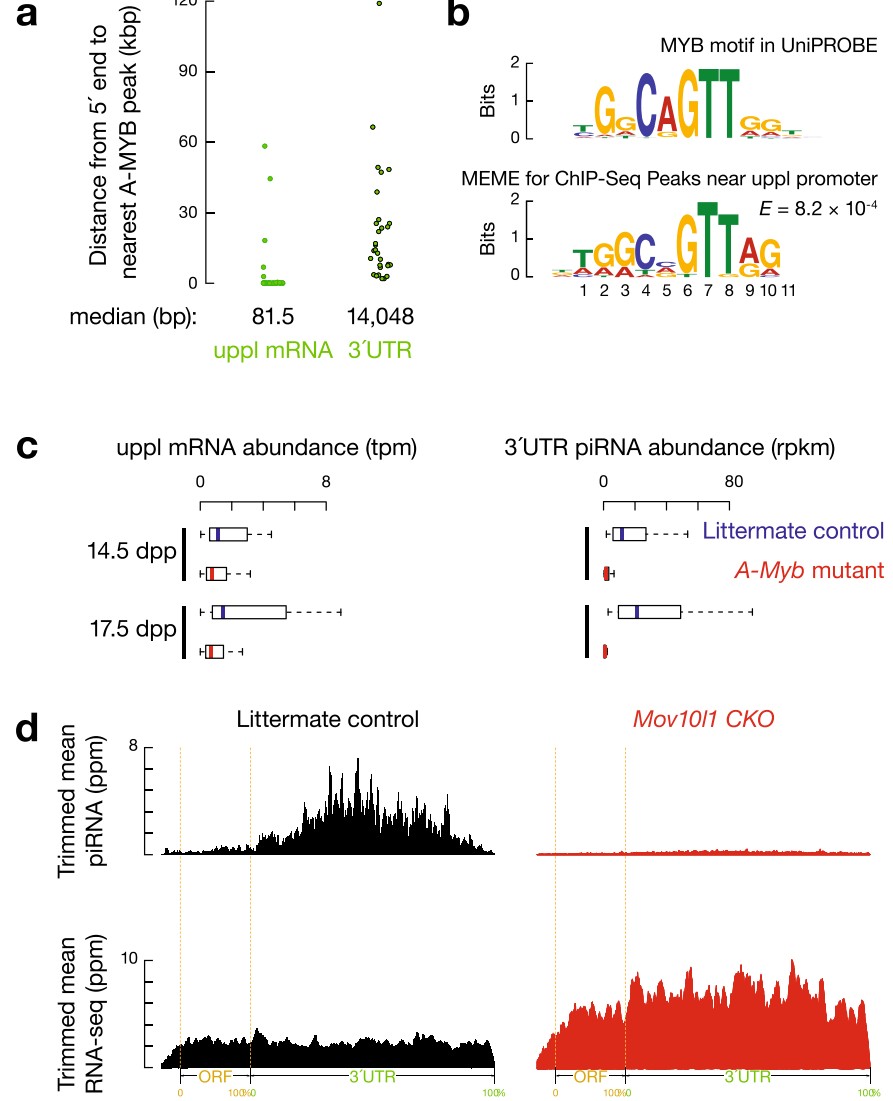

**Fig. 1 mRNA precursors of 3′UTR piRNA are regulated by A-MYB. a** The distance from the annotated transcription start site of each mRNA that comes from 3′UTR piRNA-producing loci (uppl) (left, $n = 30$) and from the 5′-ends of uppl mRNA 3′UTRs (right, $n = 30$) to the nearest A-MYB ChIP-seq peak. bp: base pair, kbp: kilobase pair. **b** Top, the MYB motif from the mouse UniPROBE database. Bottom, MEME-identified sequence motif in the A-MYB ChIP-seq peaks near the uppl mRNA transcription state sites. $E$-value computed by MEME measures the statistical significance of the motif. The information content is measured in bits and, in the case of DNA sequences, ranges from 0 to 2 bits. A position in the motif at which all nucleotides occur with equal probability has an information content of 0 bits, while a position at which only a single nucleotide can occur has an information content of 2 bits. **c** Boxplots showing the abundance of uppl mRNA (left) and 3′UTR piRNA (right) per uppl gene in *A-Myb* mutants (red) and their heterozygous littermates (blue) in testes at 14.5 and 17.5 dpp. Dpp: days postpartum, tpm: transcript per million, rpkm: reads per kilobase million. uppl $n = 30$. Box plots show the 25th and 75th percentiles, whiskers represent the 5th and 95th percentiles, and midlines show median values. **d** Aggregated data for 3′UTR piRNA abundance (upper) and uppl mRNA abundance (bottom) on uppl mRNAs (10% trimmed mean) from adult *Mov10l1*^{CKO/Δ} testes (left) and *Mov10l1*^{CKO/Δ} *Neurog3-cre* (right). Signals are aligned to the transcriptional start site (5′cap) and site of polyadenylation (3′PolyA) and further aligned to the ORF regions. Dotted lines show the translation start codon (ORF Start) and stop codon (ORF End). The x-axis shows the median length of these regions. Ppm: parts per million.

producing loci (uppl) (median distance >14 kb, Fig. 1a) but close to the TSSs of uppl mRNAs (median distance 81.5 nucleotides, nts; Fig. 1a). These sites are also enriched for MYB consensus sequences (Fig. 1b). To verify that A-MYB regulates uppl mRNAs, we studied mice carrying an *A-Myb* hypomorphic mutation[58], which displayed significantly decreased levels of uppl mRNAs based on RNA sequencing (RNA-seq) data from 14.5 dpp ($p = 4.7 \times 10^{-2}$) and 17.5 dpp testes ($p = 1.0 \times 10^{-3}$, Fig. 1c, left, and Supplementary Fig. 1b). Therefore, as it does for pachytene piRNA precursors, A-MYB directly regulates the transcription of uppl mRNAs.

To test whether uppl mRNAs are 3′UTR piRNA precursors, we examined the effect of the loss of transcription of uppl mRNAs on 3′UTR piRNA biogenesis in *A-Myb* mutant mice. Like uppl mRNAs, 3′UTR piRNA abundance significantly decreased in mutant testes (Fig. 1c, right, $p \leq 4.8 \times 10^{-10}$, and Supplementary Fig. 1b). 3′UTR piRNA depletion associated with uppl mRNA transcription loss could reflect an indirect effect of the meiotic arrest caused by the *A-Myb* mutant[58] or the lack of piRNA biogenic factors, the transcriptional activation of which requires A-MYB[8]. To test these possibilities, we blocked piRNA processing but not meiosis by conditionally knocking out (CKO) *Mov10l1* in

spermatocytes (Supplementary Fig. 1c)[11]. In the following analyses, we defined a group of non-piRNA-producing mRNAs that display similar expression dynamics in mouse testes as uppl mRNAs to be the developmentally matched-control mRNAs for comparison (Supplementary Data 1). These control mRNAs remained unchanged in *Mov10l1* CKO mutants compared to control littermates (Supplementary Fig. 1d, e). We found that 3′ UTR piRNAs are depleted in *Mov10l1* CKO mutants (Fig. 1d, upper), whereas uppl mRNAs showed significant accumulation in *Mov10l1* CKO mutants (Fig. 1d and Supplementary Fig. 1e, $p = 4.1 \times 10^{-11}$). We ruled out the possibility that the increased steady-state of uppl mRNAs is due to their transcriptional activation in *Mov10l1* CKO mutants (Supplementary Fig. 1f, $p = 0.13$). Together with the cross-linking immunoprecipitation data (CLIP)[13] that indicate MOV10L1 specifically binds to the entire length of uppl mRNAs (Supplementary Fig. 1g), our results suggest that the increased steady-state abundance of uppl mRNAs is due to the lack of 3′UTR piRNA biogenesis. Overall, our results indicate that 3′UTR piRNAs come from full-length uppl mRNAs, not from isoforms derived from uppl 3′UTRs.

**3′UTR piRNAs are derived from processed mRNAs.** A recent study suggests that unspliced transcripts are substrates for piRNA processing[59]. However, our data indicated that 3′UTR piRNAs were disproportionately produced after intron removal: >99% of piRNAs mapped to exons, compared to <0.1% that mapped to introns (17,200 ppm unique mapping 3′UTR piRNA reads from adult testes). After correcting for the length of exons and introns in piRNA-producing primary transcripts, exon-derived piRNAs were enriched by 630-fold compared to intron-derived piRNAs (Supplementary Fig. 2a). We detected piRNAs that failed to map to the genome but instead mapped to exon–exon junctions (Fig. 2a), further indicating that piRNAs were produced after intron removal and exon–exon joining. In uppl mRNA genes, 98% of the introns (207 out of 211) contained canonical GT-AG splice sites, not significantly different from the introns in control mRNA genes (326 out of 329 mRNA loci, $\chi^2$ test, $p = 0.55$), suggesting that uppl mRNAs were spliced conventionally. Furthermore, the density of piRNAs falls off sharply after the 3′end of the transcript, i.e., the site of polyadenylation (Supplementary Fig. 2a). Taken together with our previous piRNA precursor studies based on a combination of cap analysis of gene expression (CAGE) and the polyadenylation site sequencing (PAS-Seq)[8], we conclude that 3′ UTR piRNAs were produced after their precursor transcripts are fully processed—capped, spliced, and polyadenylated.

**uppl mRNA precursors harbor extensive 3′UTRs.** We used our recent reconstruction of the mouse testis transcriptome[60], combining CAGE, PAS-seq, and single-molecule long-read sequencing, to characterize the transcript structure of uppl mRNAs in comparison to that of developmentally matched and non-piRNA-producing control mRNAs. On average, mature uppl transcripts ($p < 2.2 \times 10^{-16}$), but not primary transcripts ($p = 0.18$), were longer than control mRNAs (Fig. 2b and Supplementary Fig. 2b). Given similar numbers of splicing events (no significant difference in the number of introns, $p = 0.23$, Supplementary Fig. 2c), uppl mRNAs harbored longer exons with a median length of 153 nts (Supplementary Fig. 2d; $p = 3.1 \times 10^{-14}$) versus the control mRNAs with a median exon length of 109 nts, similar to that of typical vertebrate mRNAs[61]. Since exon length distribution is associated with the relative position in the transcripts, we further separated exons into first, middle, and last exons. The last exons for the control mRNAs and uppl mRNAs are the longest among the three categories, whereas the middle exons are the shortest (Fig. 2c). While there is no significant difference in the length

distribution of the middle exons ($p = 0.65$), both the first and the last exons of uppl mRNAs are significantly longer than those of the control mRNAs ($p \le 4.2 \times 10^{-8}$, Fig. 2c). Long first exons were recently reported to be a conserved feature for pachytene piRNA precursors[40], suggesting that the unique feature of piRNA precursors can be traced back to exon–intron structures and the selection of poly-A cleavage sites.

Since 5′UTRs and 3′UTRs are typically localized to the first and last exons, respectively, we further separated the spliced transcripts according to their longest ORFs. Compared to mRNAs in other tissues, testis mRNAs are reported to have shorter 3′UTRs[62–65]. Consistent with these reports, the 3′UTRs of the control mRNAs have a median length of 392 nts. However, against the trend in testes to produce mRNAs with proximal polyadenylation sites, the 3′UTRs of the uppl mRNAs are significantly longer than those of control mRNAs, with a median length of 4583 nts (Fig. 2d; $p = 1.5 \times 10^{-11}$). While the lengths of the ORFs are similar (Fig. 2d, 1315.5 nts versus 945 nts; $p > 0.01$), the 5′UTRs of uppl mRNAs are also significantly longer than those of the control mRNAs (Fig. 2d, 346 nts versus 77 nts; $p = 3.3 \times 10^{-7}$). Therefore, regulatory sequences (UTRs), rather than ORFs, contribute to the majority of the length difference between the spliced transcripts of uppl mRNAs and control mRNAs (Fig. 2b) and render more diverse translational regulations on uppl mRNAs.

**3′UTR ribosomes guide piRNA formation.** The biogenesis of piRNAs from mRNA 3′UTRs suggests that cellular translation machinery may be used by piRNA biogenesis machinery to distinguish between 3′UTRs and ORFs. To test whether uppl mRNAs are translated, we performed ribosome profiling (Ribo-seq) in which RNA fragments protected from RNase A and T1 digestion are isolated from 80S fractions and sequenced[45]. We found that ribosomeprotected fragments (RPFs) from uppl mRNA ORFs displayed a three-nucleotide periodicity (Fig. 2e), indicating that elongating ribosomes translate these ORFs. We calculated their translational efficiency (RPF reads normalized to transcript abundance) and found that uppl mRNAs have a slightly lower translational efficiency compared to control mRNAs (Fig. 2f, 59% of the median translational efficiency of control mRNAs, $p = 0.037$), as expected given that their longer 5′ UTRs (Fig. 2c) will take longer time to scan through before the next 40S subunits start[66] and may also contain more upstream initiation sites and/or have more complex secondary structures that can reduce translational efficiency[67]. We tested codon usage, another key determination factor for translational efficiency[68], and found that uppl mRNAs and control mRNAs have a similar codon usage (Supplementary Fig. 2e, f)[69–71]. Overall, uppl mRNAs are efficiently translated on their main ORFs without obvious signs of aberrant initiation or elongation distinguishing them from control mRNAs. Taken together, uppl mRNAs function to produce both piRNAs and proteins.

To test whether ribosomes also guide piRNA biogenesis from uppl mRNAs as they do from lncRNAs[45], we identified RPF signals coming from uppl mRNA 3′UTRs (Fig. 3a). 3′UTR RPF signals were not seen in the control mRNAs (Supplementary Fig. 3a, b, $p = 1.1 \times 10^{-8}$), indicating that the distribution of RPFs on uppl mRNA 3′UTRs is RNA-specific, rather than the characteristic of typical testicular mRNAs. The in vitro RNase digestion used to obtain RPFs removed >98% of the mature 3′ UTR piRNAs in testis lysates (Supplementary Fig. 3c). We performed Ribo-seq on adult *RiboTag* mice after activating the expression of hemagglutinin (HA)-tagged RPL22 (a ribosomal large subunit protein) in germ cells[45] and affinity-purified RPFs before sequencing. After IP, sequences from mitochondrial

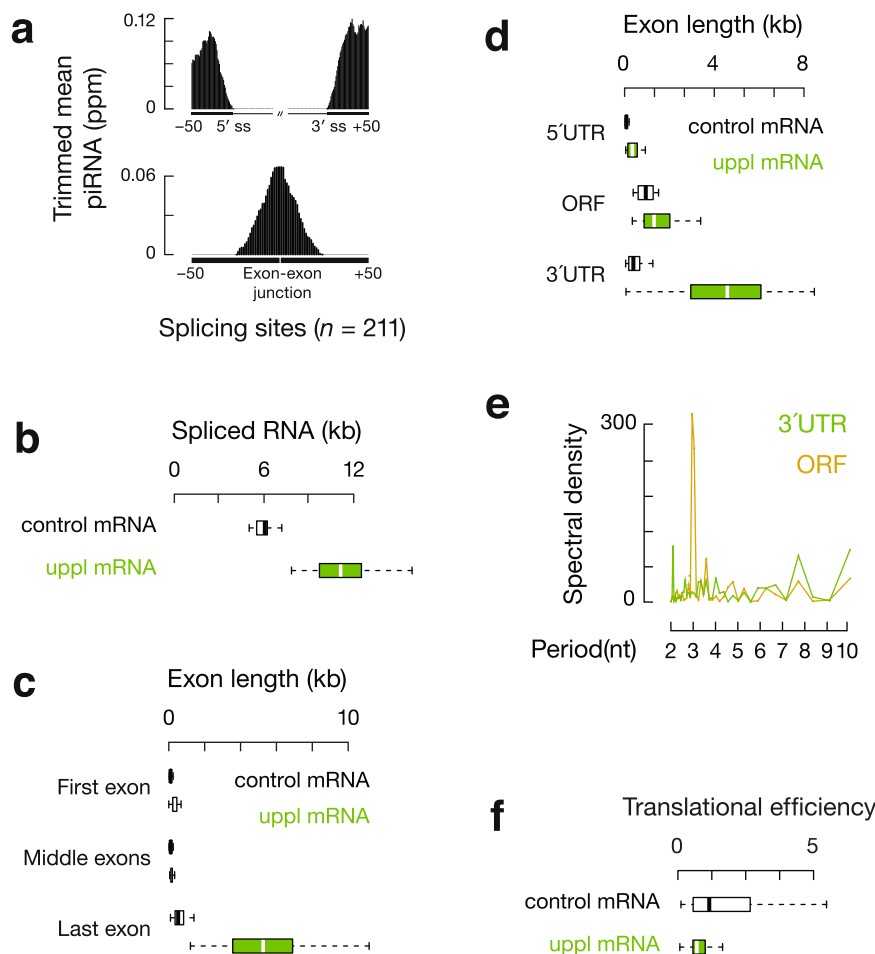

**Fig. 2 Translation and piRNA processing are coupled on uppl mRNAs. a** Top, aggregation plots of piRNA reads surrounding the 5′ and 3′ splice sites of uppl mRNAs from the adult mouse testes. Bottom, the signal was calculated for non-genome matching piRNA reads mapping to exon–exon junction sequences. **b** Boxplots showing spliced RNA transcript length distributions. Control mRNA $n = 43$, uppl mRNA $n = 30$. **c** Boxplots showing exon length distributions. Control mRNA first or last exon $n = 43$, uppl mRNA first or last exon $n = 30$. Control mRNA middle exon $n = 145$, uppl mRNA middle exon $n = 69$. **d** Boxplots showing ORF and UTR length distributions. Control mRNA $n = 43$, uppl mRNA $n = 30$. **e** Discrete Fourier transformation of the distance spectrum of 5′-ends of RPFs across ORFs (gold) and 3′UTRs (green) of uppl mRNAs in adult wild-type testes. **f** Boxplots showing translational efficiency (RPF abundancy divided by mRNA abundancy) in adult wild-type testes. Control mRNA $n = 43$, uppl mRNA $n = 30$. Box plots in (**b**–**d** and **f**) show the 25th and 75th percentiles, whiskers represent the 5th and 95th percentiles, and midlines show median values.

coding transcripts were also depleted (Supplementary Fig. 3d) given that they are translated by untagged 55S mitochondrial ribosomes. In contrast, uppl mRNA ORF and 3′UTR sequences were retained, similar to RPFs from control mRNAs (Supplementary Fig. 3d). Overall, these results indicate that the 3′UTR RPF signals are bona fide ribosomal footprints.

To test whether ribosome-bound 3′UTRs are recognized as precursors for 3′UTR piRNA production, we performed partial correlation analyses[72] between RPF abundance and piRNA abundance in uppl 3′UTRs while controlling for the abundance of uppl mRNAs, as measured by RNA-seq. These analyses can distinguish a biogenic relationship between RPFs and piRNAs or the independent correlations of RPFs and piRNAs with their uppl mRNA precursors. We found that the abundance of RPFs and piRNAs at 3′UTRs are directly correlated with each other (Fig. 3b, right, $r = 0.69$, $p = 2.9 \times 10^{-5}$). Thus, uppl 3′UTRs bound by ribosomes are processed into piRNAs.

To test whether ribosomes guide the degradation of 3′UTRs for piRNA production, we analyzed the position of the RPFs on uppl mRNA 3′UTRs. We aligned piRNAs to RPFs and plotted the 5′-ends of RPFs that overlapped with piRNAs (Fig. 3c). For RPFs, their first nucleotide overlapped with the first nucleotide of

piRNAs significantly more than with nucleotides residing 50 nts upstream or downstream (Fig. 3c, right, Z score = 36 ± 1, Z scores indicate how many standard deviations an element is from the mean; Z score > 3.3 corresponds to $p < 0.01$). We ruled out the possibility that this overlap could occur by chance or be due to ligation bias (Supplementary Fig. 3e, f). Consistent with their 5′-overlap, we found that the 5′-ends of RPFs from 3′UTRs displayed a uridine bias at the 5′-most position (1U) (Fig. 4a), reminiscent of the 1U bias in piRNAs. These results indicate that 3′UTR ribosomes dwell at the sites that represent future piRNAs.

Given that in vitro RNA digestion (RNase T1&A) used to obtain RPFs does not yield a 1U bias, we tested the possibility that these RPFs are processed in vivo by piRNA processing machinery. We modified the conventional Ribo-seq procedure (which detects both 5′P and 5′OH RPFs) to specifically capture 5′P RPF and 5′OH RPF species separately (Supplementary Fig. 4a). 5′P species are products of in vivo enzymatic cleavage, whereas the 5′-hydroxyl (5′OH) species mainly arise from in vitro RNase treatment (RNase T1&A in our procedure). To prevent mature piRNA contamination, we used the RPFs from affinity-purified 80S ribosomes for library construction (Supplementary Fig. 4a). The 5′OH RPFs showed the expected in vitro digestion signature,

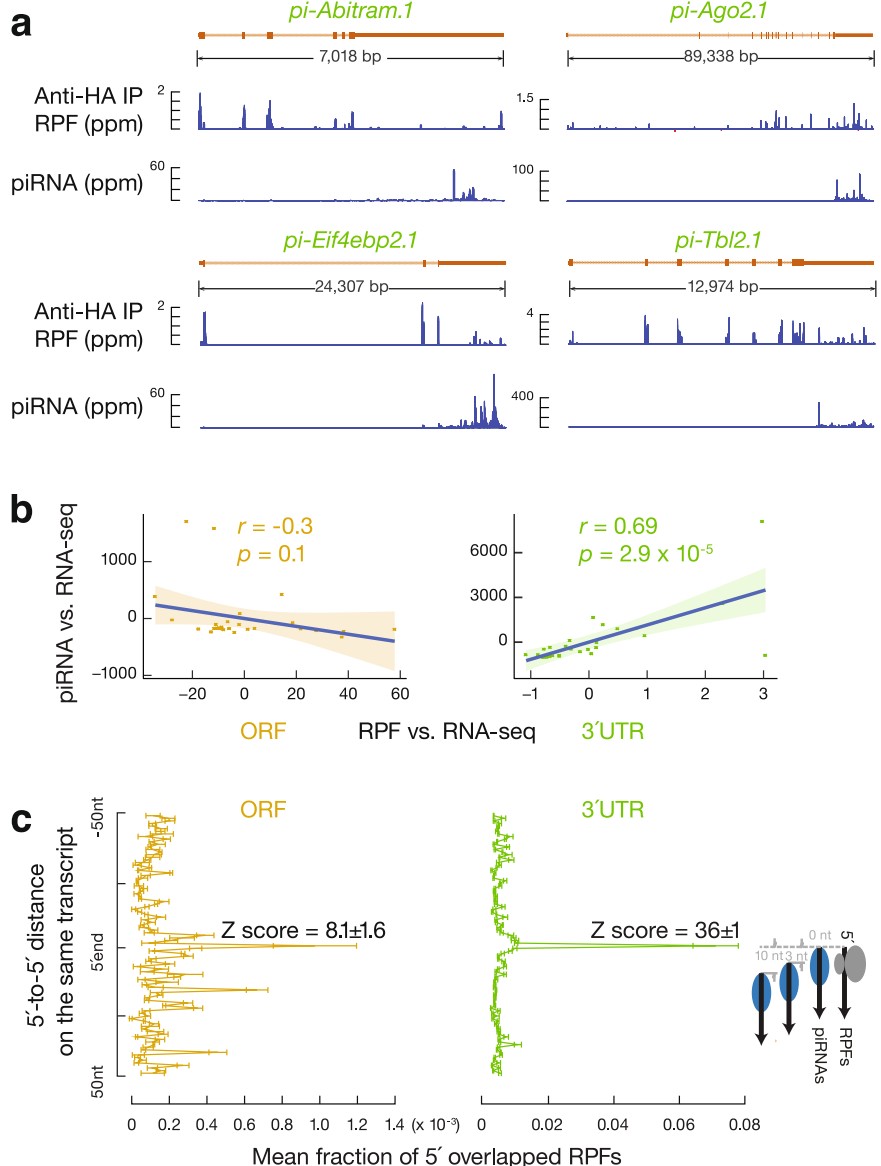

**Fig. 3 3′UTR ribosomes guide piRNA 5′-end formation. a** Normalized reads of anti-HA-immunoprecipitated RPFs and piRNAs mapping to four representative uppl genes in adult testes. **b** Residual plot of partial correlation between piRNA and RPF abundance conditional on RNA-seq abundance at the uppl ORFs (left) and 3′UTRs (right) in adult testes. Linear regression was performed between piRNA and RNA-seq abundance and between RPF and RNA-seq abundance separately, and residuals were plotted with smooth parameter "method = lm". Data are presented as mean values ± standard deviation shown as error bands. **c** Distance spectrum of 5′-ends of RPFs (26–32 nts) from ORFs (left) and from 3′UTRs (right) that overlap piRNAs in adult testes (sample size $n = 3$ independent biological replicates). Data are mean ± standard deviation.

even at uppl 3′UTRs (Supplementary Fig. 4b). However, the 5′P RPFs are >32-fold enriched relative to 5′OH RPFs at uppl 3′UTRs in comparison to uppl ORFs (Fig. 4b, $p = 1.0 \times 10^{-6}$). This indicates that RPFs predominantly present with 5′P ends in 3′ UTRs at steady state, suggesting that in vivo cleavage occurs efficiently on ribosome-bound 3′UTRs.

Consistent with these sites representing hot spots for efficient in vivo cleavage, 3′UTR ribosomes are significantly enriched in the monosome fractions relative to polysome fractions in comparison to ORFs of the control mRNAs (Fig. 4c, $p = 2.5 \times 10^{-10}$), as measured by Ribo-seq performed on purified monosome and polysome fractions as we reported previously[45]. The 1U signature of 5′P RPFs occurs specifically at the uppl 3′ UTRs but not at the ORFs of uppl RNA or the ORFs of control mRNAs (Supplementary Fig. 4b), indicating that 3′UTR RPFs are cleaved by the piRNA processing machinery. Altogether, piRNA

processing machinery generates the 5′P in vivo cleavage products with a ribosome bound at their 5′ extremities, and these 5′P ends become the 5′-ends of future piRNAs.

**3′UTR piRNA biogenesis is coupled with upstream translation.** The translation of uppl mRNAs could either be coupled or uncoupled with piRNA biogenesis. If coupled, there would be a limited time window for translation to occur before the mRNA can be cleaved to generate piRNAs. If uncoupled, uppl mRNAs should undergo multiple rounds of translation. In mammals, two types of cap-binding proteins (CBPs) participate in protein synthesis[73]. The largely nuclear CBPs 80 and 20 (CBP80/20), which mediate the pioneer round of translation once newly synthesized mRNAs reach the cytoplasm[74,75], are ultimately replaced by the major cytoplasmic CBP eukaryotic translation

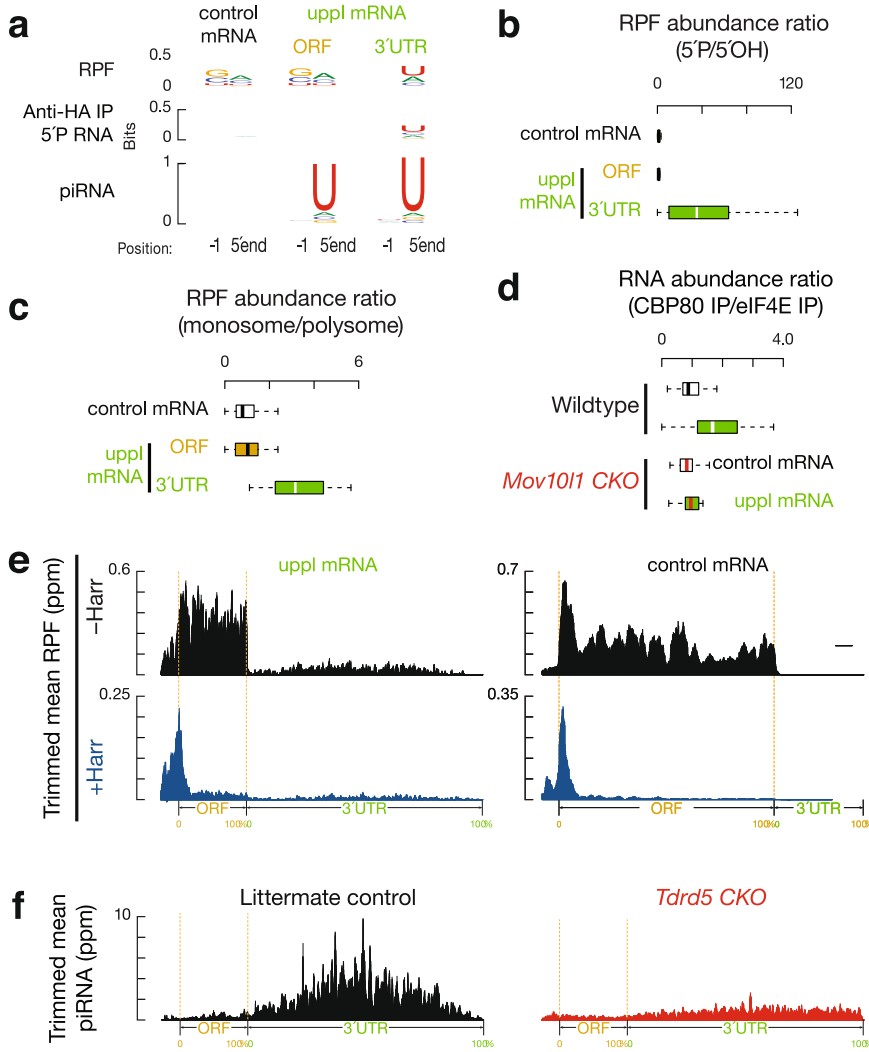

**Fig. 4 Biphasic piRNA biogenesis. a** Sequence logos depicting nucleotide (nt) bias at 5'-ends and 1 nt upstream of 5'-ends. Top to bottom: RPF species, anti-HA-immunoprecipitated 5'P RNA species, and piRNAs in adult testes, which map to mRNAs, uppl mRNA ORFs, and 3'UTRs, respectively. **b** Boxplots of the ratios of 5'P RPF versus 5'OH RPFs in adult wild-type testes. Control mRNA $n = 43$, uppl mRNA $n = 30$. **c** Boxplots of the ratios of RPF abundance in monosome fractions versus those in polysome fractions in adult wild-type testes. Control mRNA $n = 43$, uppl mRNA $n = 30$. **d** Boxplots of the ratios of RNA abundance in CBP80 IP versus eIF4E IP in adult wild-type (upper) and *Mov10l1* CKO mutant (lower) testes. Control mRNA $n = 43$, uppl mRNA $n = 30$. **e** Aggregated data for RPF abundance from untreated adult testes (top) and from harringtonine-treated adult testes (bottom) across 5'UTRs, ORFs, and 3'UTRs of the uppl mRNAs (left) and control mRNAs (right). The x-axis shows the median length of these regions, and the y-axis represents the 10% trimmed mean of relative abundance. **f** piRNA abundance on uppl mRNAs (10% trimmed mean) in adult testes. Control (left) and *Tdrd5* CKO (right). Box plots in (**b–d**) show the 25th and 75th percentiles, whiskers represent the 5th and 95th percentiles, and midlines show median values.

initiation factor 4E (eIF4E)[76]. To detect the nature of these CBPs during the translation of mRNAs that generate piRNAs, we immunoprecipitated (IP) CBP80-bound RNAs and eIF4E-bound RNAs and subjected them to RNA-seq. We calculated RNA abundance after CBP80 IP versus eIF4E IP and found that uppl mRNAs are significantly enriched in the CBP80 IP in comparison to control mRNAs (Fig. 4d, upper, $p = 1.3 \times 10^{-5}$). To test whether the enrichment of uppl mRNA with CBP80 is due to piRNA processing, we repeated the CBP80 IPs and eIF4E IPs using *Mov10l1* CKO mutants. The enrichment of uppl mRNAs in CBP80 IPs was significantly reduced in *Mov10l1* CKO mutants compared to the wild type (Fig. 4d, $p = 3.7 \times 10^{-4}$), while the distribution of control mRNAs was unaffected ($p = 0.32$). Our results indicate that our failure to detect appreciable translation of eIF4E-bound uppl mRNAs is due to their processing into piRNAs while they are CBP80-bound, suggesting that uppl mRNA

translation and piRNA biogenesis are temporally coupled and carried out using the same uppl RNA molecule.

The coupling of piRNA processing and translation indicates that the uppl mRNAs are targeted by the piRNA processing machinery immediately after or while they are being translated. To investigate how this coupling is achieved, we tested whether 3'UTR ribosome binding requires upstream initiation near the 5'-cap. We treated testes with harringtonine[45], which blocks early elongation by binding to the A site of newly assembled ribosomes[77]. Given that uppl 3'UTRs do not harbor long ORFs (median putative ORF length is 42 nts, Supplementary Fig. 4c), 3'UTR ribosomes should accumulate and be resistant to harringtonine treatment if there is internal initiation. After a 2-h treatment, the control mRNAs showed a significant reduction in elongating ribosomes on ORFs ($p = 8.7 \times 10^{-12}$, Fig. 4e and Supplementary Fig. 4d), compared to the initiating ribosomes accumulated

around the start codons. On uppl mRNAs, ribosomes accumulated at the translation start sites, similar to control mRNA ($p = 0.17$). In comparison to the ribosomes at translation start sites, ribosomes at ORFs ($p = 4.4 \times 10^{-11}$) and 3′UTRs ($p = 1.2 \times 10^{-11}$) were substantially reduced (Fig. 4e and Supplementary Fig. 4d). Thus, 3′UTR ribosomes migrate from upstream long ORFs rather than loading internally. Unlike the conventional initiation mechanisms on mRNA ORFs after uORF that require a short uORF length, our study suggests compromised post-termination recycling, which underlies the coupling between translation at ORFs and piRNA processing at 3′UTRs.

**Biphasic biogenesis before and after the stop codon**. uppl ORFs also produce authentic piRNAs with a 1U bias (Fig. 4a), although piRNAs derived from 3′UTRs are >31-fold more abundant than they are in ORFs, which equates to an 8-fold difference when normalized to 3′UTR and ORF length, respectively (Supplementary Fig. 4e). To understand why significantly different amounts of piRNAs are detected from ORFs and 3′UTRs ($p = 1.4 \times 10^{-7}$, Supplementary Fig. 4e) and how this is related to ribosomes, we analyzed uppl ORF RPFs. Unlike ORF piRNAs with a 1U bias, ORF RPFs exhibit an in vitro digestion signature without a 1U bias (Fig. 4a). The ratios of 5′P RPFs to 5′OH RPFs from uppl ORFs are comparable to those of RPFs from the control mRNA ORFs (Fig. 4b, $p = 0.71$), suggesting that RPFs present at uppl ORFs predominately harbor 5′OH ends at the steady state. RPFs from uppl ORFs have a significantly less pronounced 5′ overlap with mature piRNAs than 3′UTR RPFs (Fig. 3c, Student's t test, $p = 1.4 \times 10^{-4}$). Consistent with the lack of in vivo processing of ORF RPFs, we found that ORF RPFs have a similar distribution between monosome and polysome fractions when compared to RPFs from the control mRNA ORFs (Fig. 4c, $p = 0.45$). Furthermore, the abundance of RPFs and piRNAs at ORFs do not correlate with each other when controlling for the abundance of uppl mRNAs (Fig. 3b, left, $r = -0.3$, $p = 0.1$), indicating a lack of a biogenic relationship between ribosome-bound ORFs and piRNAs. Therefore, our results indicate that ribosome-guided piRNA processing occurs at uppl 3′UTRs but not at uppl ORFs.

The lack of a biogenic relationship between RPFs and piRNAs at uppl ORFs could be due to: (1) a different mechanism of piRNA processing where only a translationally suppressed subpopulation of uppl mRNA ORFs is processed into piRNAs, with the majority protected from processing; or (2) the mRNA ORFs are processed but do not generate piRNAs efficiently. To distinguish between these two possibilities, we monitored the fragmentation process on ribosomes. Using degradome sequencing (degradome-seq) of long RNAs (>200 nt) from affinity-purified ribosomes, we captured ribosome-bound 5′P RNAs. We found that the abundance of ribosome-bound 5′P RNAs from 3′UTRs was not higher than that from ORFs ($p = 0.3$, Supplementary Fig. 4e), arguing against the possibility that the majority of ORFs were protected from processing. We then tested whether these 5′P RNAs have their 5′-ends aligned with the 5′-ends of piRNAs, which would suggest that the cleavage process forms the piRNA 5′-ends. We found that the 5′P RNAs from uppl ORFs had a significantly lower 5′ overlap (Z score = 3.2 ± 0.4) with the 5′-ends of uppl ORF piRNAs compared to the 5′ overlap between 5′P RNAs and piRNAs at uppl 3′UTRs (Z score = 18 ± 1, Student's t test, $p = 1.8 \times 10^{-5}$, Supplementary Fig. 4f). Consistent with this, neither ribosome-bound 5′P RNAs nor RPFs from uppl ORFs have the 1U bias, unlike that of the 5′P RNAs and RPFs from uppl 3′UTRs (Fig. 4a). Thus, although uppl ORFs are cleaved, the lack of 5′ overlap between the cleavage products and uppl ORF piRNAs indicates that the cleavage products are inefficiently processed into piRNAs.

To determine whether auxiliary factors facilitate efficient processing of piRNAs at uppl 3′UTRs, we tested for TDRD5 (tudor domain containing 5) function, the disruption of which impacts pachytene piRNA production[45,78]. We found that piRNAs derived from uppl ORFs were still produced, but 3′UTR piRNAs were depleted in *Tdrd5* mutants (Fig. 4f and Supplementary Fig. 4f), indicating that uppl ORF piRNAs do not require TDRD5 for their production, whereas 3′UTR piRNAs do. Therefore, 3′UTR piRNAs derived from uppl 3′UTRs and ORFs have linked but distinct biogenic requirements. Taken together, our results indicate that efficient piRNA processing at uppl 3′UTRs requires both ribosomes and TDRD5 and that post-transcriptional processing differences at 3′UTRs and ORFs explain why even though the entire length of uppl mRNAs generates piRNAs, >96% of uppl piRNAs are derived from uppl 3′UTRs. These results are similar to the biphasic piRNA biogenic mechanism we identified previously in pachytene piRNAs from lncRNA piRNA precursors[45]. Therefore, combined with our previous work on lncRNAs[45], our study reveals a general ribosome-guided mechanism by which piRNA precursors, regardless of their source, are converted into piRNA sequences (Supplementary Fig. 4g).

**Inhibition of co-translational surveillance pathways**. Ribosomes that failed to recycle at the 3′UTRs of mRNAs should be rescued/resolved by ribosome recycling factors such as PELOTA (the mouse homolog of yeast DOM34)[79–84]. To understand how 3′UTR ribosomes avoid recycling by PELOTA, we performed immunostaining on squashed spermatocytes and spermatids. We found that PELOTA localized to the nuclei of pachytene spermatocytes, but not to the nuclei of round spermatids (Fig. 5a). Similar nuclear localization, specifically at the pachytene stage, has been reported for other translational control proteins[85]. Considering that PELOTA is the major player in no-go decay[86] and no-stop decay[87], the sequestration of ribosome recycling factors in the nucleus suggests that their associated mRNA decay pathways are inhibited at pachynema.

Although uppl mRNAs that have extensive 3′UTRs (Fig. 2d) should represent conventional substrates to elicit NMD[88], the spreading of ribosomes on 3′UTRs should inhibit the NMD as demonstrated by polycistronic viruses or inhibition of ribosome recycling factors[89,90]. To test this idea, we performed IP in testis lysates using anti-UPF1 and anti-phosphorylated UPF1, followed by RNA-seq. UPF1 plays a role in NMD target recognition and elimination, and the phosphorylation of UPF1 is required to activate NMD[74,91]. We found that uppl mRNAs are enriched in UPF1 IP (Fig. 5b, $p = 3.9 \times 10^{-8}$), but this uppl mRNA-associated UPF1 did not show increased levels of phosphorylation compared to control mRNAs (Fig. 4b, lower, $p = 0.15$), indicating the NMD is not activated on the uppl mRNAs given that they appear to be NMD substrates. The NMD pathway is further repressed globally at the pachynema in testes[92]. The lack of NMD activity, in conjunction with the nuclear localization of PELOTA/DOM34 at the pachytene stage, may be a prerequisite for piRNA biogenesis during normal development. Taken together, ribosomes binding to the piRNA precursor uppl mRNAs temporally stagger with other translation-dependent mRNA decay pathways, allowing massive piRNA production from mRNAs and lncRNA precursors during the pachytene stage with an abundance of 3.8–8.4 million piRNA molecules in each spermatocyte[25] constituting >95% of the total piRNAs in adult mouse testes[8].

**piRNA biogenesis fine-tunes protein production**. Given that other translation-dependent mRNA decay targets endogenous mRNAs to fine-tune protein production[93], we tested whether

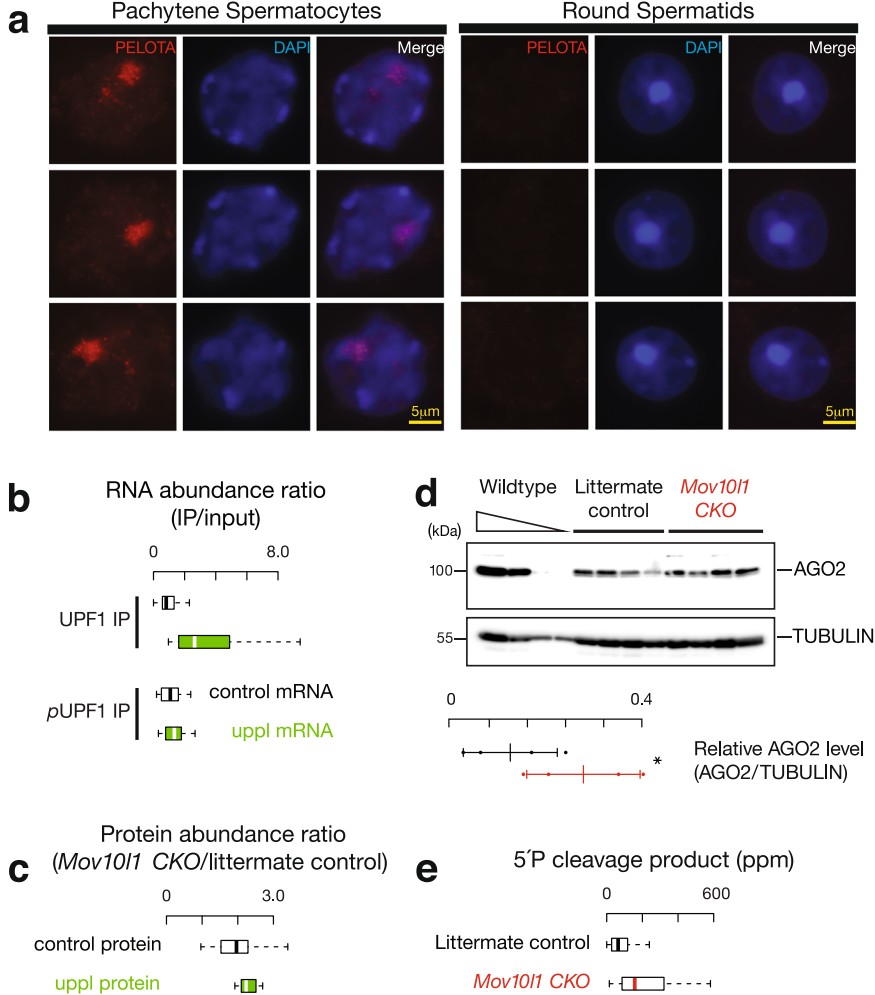

**Fig. 5 3′UTR piRNA biogenesis as a special co-translational mRNA decay pathway. a** Immunolabeling of squashed pachytene spermatocytes (left) and round spermatids (right). Three examples of each stage were shown using anti-PELOTA, DAPI, and merged. Experiments were repeated at least three times independently with similar results. Scale bar, 5 μm. **b** Boxplots of the ratios of RNA abundance in UPF1 IP (upper) and phosphorylated UPF1 IP mutant (lower) versus their corresponding input in adult wild-type testes. Control mRNA $n = 43$, uppl mRNA $n = 30$. **c** Boxplots of the changes of protein abundance coded per gene in *Mov10l1* mutants (sample size $n = 5$ independent biological replicates) compared to littermate controls in adult testes. Control mRNA $n = 43$, uppl mRNA $n = 30$. **d** AGO2 western blot (top) and TUBULIN western blot (bottom) of adult testis lysates from serial dilutions of adult wild-type lysates (left) and four biological replicates of *Mov10l1*$^{CKO/\Delta}$ *Neurog3-cre* (middle) and *Mov10l1*$^{CKO/\Delta}$ (right). Abundance of AGO2 and TUBULIN was quantified using ImageJ[158] and then fitted to the standard curve of the wild-type testis lysates. Relative abundance of AGO2 were calculated as the abundance of AGO2 normalized by the abundance of TUBULIN (sample size $n = 4$ independent biological samples). Data are mean ± standard deviation; *$p = 0.048$, one-side Student's $t$ test. Each data point was overlaid as dot plots. **e** Boxplots of the 5′P decay intermediates per mRNA in adult wild-type testes. AGO2 target transcripts $n = 41$. Box plots in (**b**, **c**, **e**) show the 25th and 75th percentiles, whiskers represent the 5th and 95th percentiles, and midlines show median values.

the piRNA biogenesis pathway impacts protein production from uppl mRNAs. We performed quantitative mass spectrometry using *Mov10l1* CKO mutants and their littermate controls. We observed a significant increase in steady-state protein levels from uppl mRNAs, with a median increase of 13%, in comparison to the control mRNAs ($p = 3.2 \times 10^{-2}$, Fig. 5c). These results indicate that piRNA biogenesis regulates the abundance of a subset of proteins, including AGO2 (encoded by *pi-Ago2/Eif2c2.1*), which plays a central role in small RNA function, DDX19B (encoded by *pi-Ddx19b.1*, the homolog of yeast *Dbp5*), which controls mRNA export[94,95], and the translational repressor 4E-BP1 (encoded by *pi-Eif4ebp2.1*)[96]. Among the 16 out of 30 uppl genes with reported mutant phenotypes[97], half of them have been demonstrated to be essential, 5 of which (*Ddx19b*, *Strbp*, *Asb1*, *Ip6k1*, and *Nr2c2*) exhibit impaired male fertility[98–102] and 3 of which (*Ago2*,

*Exoc8*, and *Ipmk*) display embryonic lethality[98,103,104]. This is significantly different ($\chi^2$ test, $p = 1.3 \times 10^{-4}$) from the reported phenotypes in all mouse mutants wherein 7% and 30% of mutants display fertility defects and embryonic lethality, respectively[105]. Thus, 3′UTR piRNA processing affects the level of proteins from essential genes.

To test the impact of protein amount changes due to 3′UTR piRNA biogenesis, we chose to focus on the miRNA function of AGO2, whose abundance significantly increased by 115% according to western blot quantification (Student's $t$ test, $n = 4$, $p < 0.05$, Fig. 5d) in total testes of *Mov10l1* CKO mutants where piRNA biogenesis is blocked. We focused on miRNAs that are regulated by A-MYB to ensure that the targeted events occur at the same developmental stage when 3′UTR piRNAs are produced. We identified six miRNAs (mmu-mir-449a,c, mmu-mir-34b,c, mmu-mir-184, and mmu-mir-191) whose expression are

significantly reduced in the *A-Myb* mutant ($q$ value < 0.05) and have an A-MYB binding peak nearby (within 1 kb) (Supplementary Fig. 5a, b). Given that miRNAs can be loaded into all four AGO proteins (AGO1–4), to distinguish the function of AGO2-bound miRNAs, we took advantage of the fact that only AGO2 has slicer activity[106] and searched for the targets of miRNA-guided AGO2-mediated cleavage with the following criteria: (1) the cleavage products are detected in a degradome-seq library that captures the 5'P species in adult testis; and (2) the target RNAs are directly regulated by A-MYB to ensure that the targeted events occur at the same cell type and same developmental stage when 3'UTR piRNAs are produced. We detected 41 target transcripts meeting these criteria. The number of cleavage events in these transcripts, as measured by degradome-seq, significantly increased with a median increase of 146% in *Mov10l1* CKO mutants (Fig. 5e, $p = 2.5 \times 10^{-4}$). In sum, the increased protein level of AGO2 in *Mov10l1* CKO mutants leads to increased activity of miRNAs expressed during pachynema.

Among the 17 out of 41 target transcripts of miRNA-guided AGO2-mediated cleavage with reported mutant phenotypes[97], 14 (>80%) have been demonstrated to be essential, 4 (*Atp8b3*, *Btrc*, *Cfap206*, and *Ptdss2*) exhibit impaired male fertility[107–110], and 10 (*Dnaaf1*, *Eif3e*, *Ipo11*, *Ndufaf7*, *Rpa1*, *Slc2a3*, *Smarcb1*, *Smc5*, *Ssrp1*, and *Tmem258*) display embryonic lethality[98,111–115]. *Rpa1* (replication protein A1) and *Slc2a3* (solute carrier family 2, member 3, also known as *Glut3*) are haploinsufficient[116,117], indicating the essence of sufficient dosage of gene products for their normal functions. As expected with increased miRNA-guided AGO2-mediated cleavage, we detected significantly decreased RPF abundance from these target RNAs in comparison to the control mRNAs with a median decrease of 29% (Supplementary Fig. 5c, $p = 1.8 \times 10^{-5}$). Given the sensitivity to gene dosage and the essential functions of these target genes, increased miRNA-guided AGO2-mediated cleavage of their transcripts and decreased protein synthesis may contribute to the infertility of *Mov10l1* CKO mutants. Considering that AGO2 is just one of the uppl mRNA protein products, our data support the biological significance of 3'UTR piRNA biogenesis in fine-tuning protein abundance during normal development.

**The biogenesis of 3'UTR piRNAs is evolutionarily conserved.** To test whether the biogenic mechanisms for 3'UTR piRNAs are also seen for 3'UTR piRNAs found in other amniotes, we identified and annotated 3'UTR piRNAs in roosters (*Gallus gallus*). We used RNA-seq from roosters to assemble the testis-specific transcriptome and then aligned piRNAs to annotated mRNAs. To identify precursor transcripts, we required a piRNA abundance of >100 ppm and ≥90% of piRNAs mapping to 3'UTRs (a median percentage of 96.3% of mouse 3'UTR piRNAs derived from 3'UTRs). To ensure that mRNAs are translated in rooster testes, we also required an RPF abundance ≥1 ppm from their ORFs. Using these criteria, we detected, in total, 23 transcripts that both produce piRNAs (Fig. 6a) and code for proteins (as shown by 3-nucleotide periodicity, Fig. 6b), thus representing uppl mRNAs. The TSSs of these transcripts (but not their 3'UTRs) have a nearby H3K4me3 ChIP-seq peak, a signature of RNA Pol II TSSs[118] (Fig. 6a), arguing against the existence of 3'UTR-specific isoforms, and 21 out of 23 H3K4me3 ChIP-seq peaks completely overlap with A-MYB ChIP-seq peaks (Supplementary Fig. 6a), indicating that 3'UTR piRNAs also exist in chickens and are derived from full-length mRNA precursors.

The 23 chicken uppl mRNAs include genes such as *pi-CRTC1* (CREB regulated transcription coactivator 1), *pi-DOT1L* (DOR1-like, histone H3 methyltransferase), and *pi-USP53* (ubiquitin specific peptidase 53). All 23 genes have mouse homologs, and

mouse mutants for 17 of the 23 have been reported[97]. Out of the 17 genes, 10 (59%) are essential in mice, with 3 impairing fertility and the rest causing embryonic lethality. Among the 30 mouse uppl genes, 23 have a homolog in chickens, none of which robustly produce piRNAs in rooster testes (Supplementary Fig. 6b). The 23 chicken uppl genes have no overlap with the mouse uppl genes, and their mouse homologs produce few piRNAs in adult mouse testes (Supplementary Fig. 6c). We used these 23 chicken transcripts as non-piRNA-producing control mRNAs in the following analyses in chickens. While none of the mouse uppl mRNAs come from sex chromosomes (X or Y chromosomes), explained by meiotic sex chromosome inactivation, 8 out of 23 (35%) of the chicken uppl genes mapped to the Z chromosomes (the bird sex chromosomes that are not inactivated because rooster is the homogametic sex), which are significantly enriched compared to all the mRNA-encoding genes we assembled for rooster testes (5.7%, $\chi^2$ test, $p = 4.0 \times 10^{-6}$). Thus, mouse and chicken 3'UTR piRNAs are derived from diverse, non-overlapping genes, the majority of which are essential for viability and fertility.

We found that RPFs also extended into the 3'UTRs of chicken uppl mRNAs (Fig. 6a). After controlling for the mRNA levels measured by RNA-seq, we found a significant partial correlation between piRNA abundance and RPF abundance from each chicken uppl mRNA 3'UTR ($r = 0.96$, $p = 1.5 \times 10^{-12}$, Supplementary Fig. 6d, left), indicating that the ribosome-bound uppl 3'UTRs are processed into piRNAs in roosters. The 5'-ends of RPFs from chicken uppl 3'UTRs significantly overlapped with the 5'-ends of piRNAs (Fig. 6c, right, and Supplementary Fig. 6e). The 5'-ends of 3'UTR RPFs also displayed a 1U bias (Fig. 6d). Authentic piRNAs with 1U bias are produced from uppl ORFs (Fig. 6d). Unlike 3'UTR RPFs, the abundance of uppl ORF RPFs does not correlate with piRNA abundance ($r = -0.09$, $p = 0.69$, Supplementary Fig. 6d, right), and the ORF RPFs did not display a signature of in vivo cleavage (Fig. 6d) nor correspond to future piRNA sites (Fig. 6c, left). Thus, chicken uppl ribosomes also guide endonucleolytic cleavages that generate piRNA 5'-ends in 3'UTRs, but do not do so in ORFs. In sum, although the mRNAs that produce piRNAs do not overlap between mice and chickens, the existence of ribosome-guided piRNA biogenesis from mRNA 3'UTRs in both mice and chickens suggests that an evolutionary conserved biogenic mechanism predates the divergence of mice and chickens approximately 330 million years ago[119].

**Transposon fragments are embedded in precursor mRNAs.** To determine the common features of 3'UTR piRNA precursors in chicken and mice, we revisited the debate over whether 3'UTR piRNA precursors harbor TE sequences[56,120,121]. We tested whether mouse uppl mRNAs are enriched in TE sequences or are produced from regions with TEs nearby when compared to the control mRNAs. We determined the distance from the start site of each transcript to the nearest TE in the genome and found that uppl mRNAs are not significantly closer to TEs than the control mRNAs (Supplementary Fig. 7a, $p \geq 0.20$). We computed the percentages of exonic and intronic nucleotides that are annotated as part of a TE for each locus and found that a significantly higher fraction of uppl mRNA exons harbor short interspersed nuclear elements (SINEs; $p \leq 2.7 \times 10^{-8}$) in comparison to the control mRNA genes that contained no exonic nucleotides corresponding to TEs (Fig. 7a). The intronic regions of uppl mRNA genes and the control mRNA genes contained a similar fraction of TEs ($p \geq 0.02$, Supplementary Fig. 7b). These SINE-containing exonic regions almost exclusively correspond to the 3'UTRs (Supplementary Fig. 7c). Thus, TEs are enriched in spliced uppl mRNAs in mice.

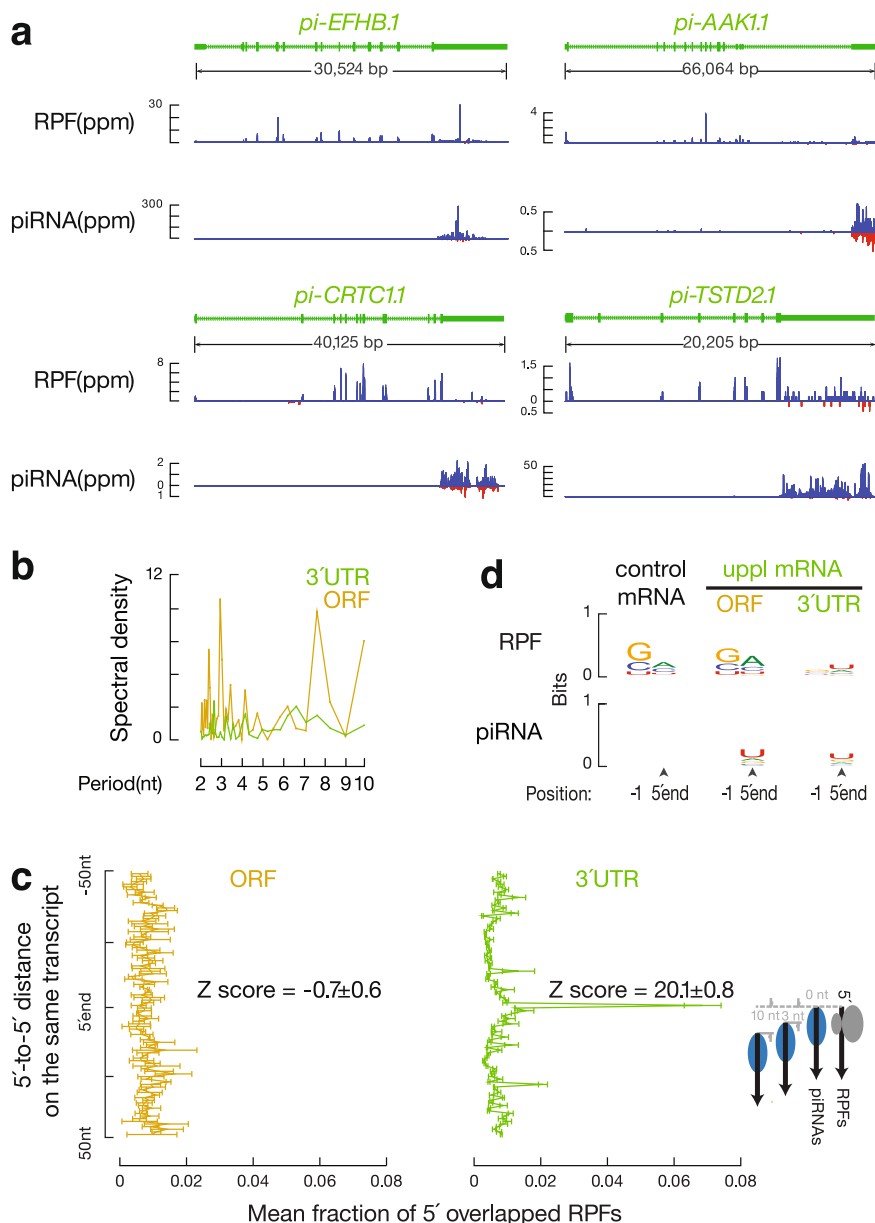

**Fig. 6 Ribosome-guided 3′UTR piRNA formation is evolutionarily conserved. a** Normalized reads of RPFs and piRNAs mapping to four representative chicken uppl genes in adult rooster testes. In roosters, both strands of uppl mRNAs code for piRNAs; blue represents Watson strand mapping reads; red represents Crick strand mapping reads. **b** Discrete Fourier transformation of the distance spectrum of 5′-ends of RPFs across ORFs (gold) and 3′UTRs (green) of chicken uppl mRNAs in adult rooster testes. **c** Distance spectrum of 5′-ends of RPFs from uppl ORFs (left) and from 3′UTRs (right) that overlap piRNAs in adult rooster testes (sample size $n = 3$ independent biological replicates). Data are mean ± standard deviation. **d** Sequence logos depicting nucleotide bias at 5′-ends and 1 nt upstream of 5′-ends of the following species from adult rooster testes. Top to bottom: RPF species and piRNAs, which map to control mRNAs, uppl ORFs, and 3′UTRs, respectively.

In adult mouse testes, we detected TE piRNAs produced from uppl mRNAs (Fig. 7b) that uniquely map to SINEs embedded in uppl mRNA 3′UTRs. Both sense and antisense piRNAs (according to their orientation with SINEs) exhibited a 1U bias not a 10A bias (Fig. 7b and Supplementary Fig. 7d), displaying a signature of primary piRNAs rather than secondary piRNAs. To test whether these piRNAs could trigger ping-pong amplification cycles in trans, we searched for secondary piRNAs from total SINE piRNAs in adult testis (piRNAs mapping to consensus SINE sequences with up to three mismatches allowed throughout the piRNA sequences). We found that antisense SINE-piRNAs, but not sense SINE-piRNAs, produced from uppl mRNAs display a significant ping-pong signature with total SINE-piRNAs (Fig. 7c

and Supplementary Fig. 7e), indicating that these 3′UTR piRNAs can trigger post-transcriptional cleavage of SINE transcripts. To test the biological significance of why only SINEs but not other TE superfamilies are embedded in mouse uppl mRNAs, which cannot be explained by their percentage in the mouse genome (8.22% is SINE, 19.20% is long interspersed nuclear element (LINE), 9.87% is long terminal repeat (LTR), and 0.88% is DNA transposons)[122], we compared their expression around the developmental stage when uppl piRNAs are produced. We found that SINEs are more highly expressed compared to DNA, LINE, and LTR transposons (Fig. 7d). Therefore, our data indicate that a subset of piRNAs produced from mouse uppl mRNAs post-transcriptionally silence TEs.

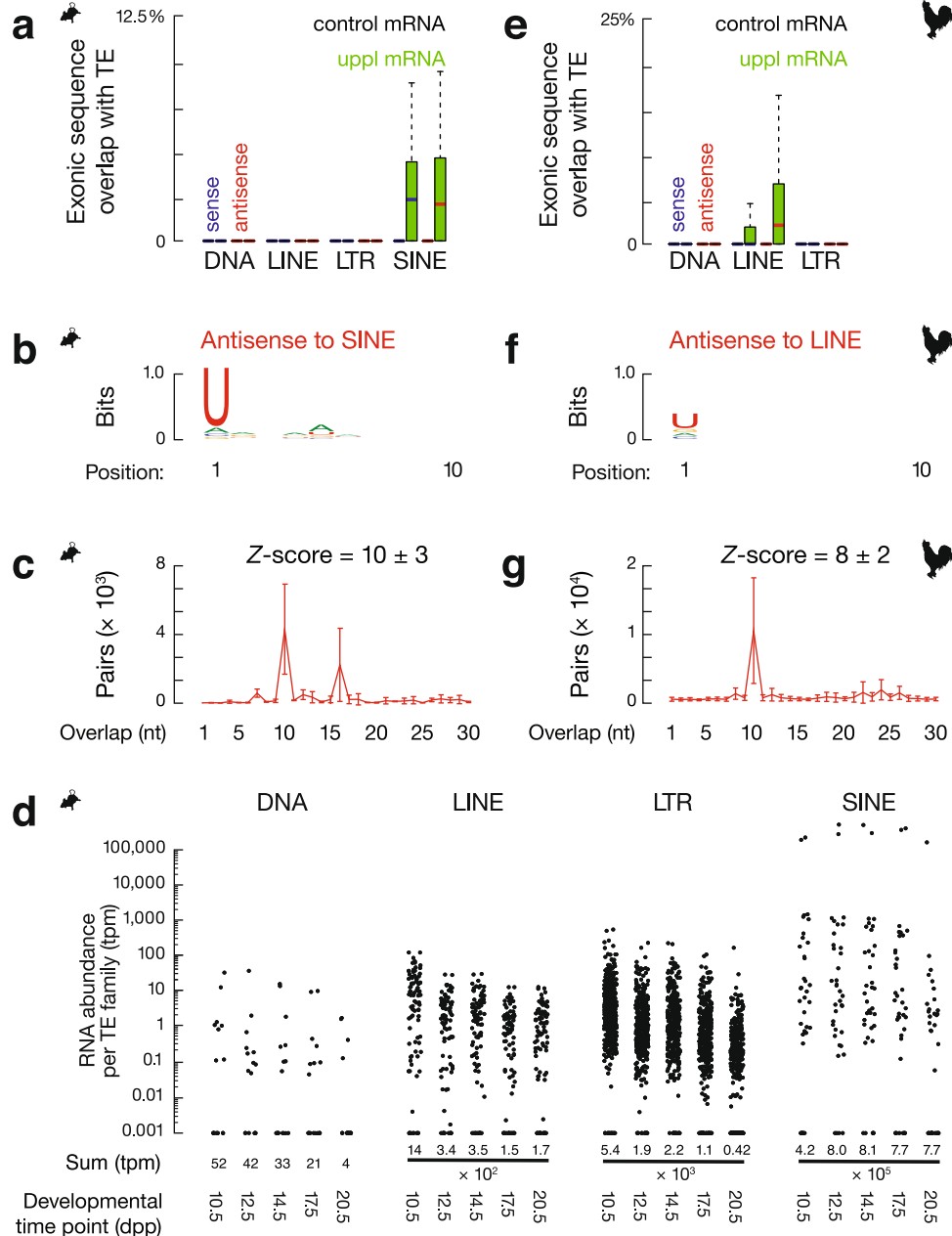

**Fig. 7 Spliced piRNA precursor mRNAs contain transposon fragments. a** Boxplots showing the fraction of transcript exon sequence correspondence to sense (blue) and antisense (red) transposon sequences in mouse genome. Control mRNA $n = 43$, uppl mRNA $n = 30$. **b** Sequence logo showing the nucleotide composition of antisense SINE-piRNA species that uniquely map to mouse uppl mRNAs. **c** The 5′–5′ overlap between piRNAs from opposite strands of SINE consensus sequences was analyzed to determine whether antisense SINE piRNAs from mouse uppl mRNAs display Ping-Pong amplification in trans. The number of pairs of piRNA reads at each position is reported (sample size $n = 3$ independent biological replicates). Data are mean ± standard deviation. The $Z$ score indicates that a significant ten-nucleotide overlap (Ping-Pong) was detected. $Z$ score $= 1.96$ corresponds to $p$ value $= 0.025$. **d** The RNA abundance of each TE superfamily in testes at the five developmental time points. Dpp: days postpartum, tpm: transcript per million. **e** Boxplots showing the fraction of transcript exon sequence correspondence to sense (blue) and antisense (red) transposon sequences in chicken genome. Control mRNA $n = 23$, uppl mRNA $n = 23$. **f** Sequence logo showing the nucleotide composition of antisense LINE-piRNA species that uniquely map to chicken uppl mRNAs. **g** The 5′–5′ overlap between piRNAs from opposite strands of LINE consensus sequences was analyzed to determine whether antisense LINE-piRNAs from chicken uppl mRNAs display Ping-Pong amplification in trans (sample size $n = 3$ independent biological replicates). Data are mean ± standard deviation. $Z$ score $= 1.96$ corresponds to $p$ value $= 0.025$. Box plots in **a**, **e** show the 25th and 75th percentiles, whiskers represent the 5th and 95th percentiles, and midlines show median values.

To test whether the function of producing antisense TE piRNAs is biologically significant during evolution, we performed a similar analysis in chickens and found that the processing of chicken 3′UTR piRNA precursors also generates antisense TE piRNAs. Chicken piRNA precursor mRNAs harbor TEs in their exonic regions that correspond to 3′UTRs (Fig. 7e and Supplementary Fig. 7f). While SINEs are largely absent from the chicken genome[123], we found significantly more LINEs embedded in chicken uppl mRNA exons ($p = 2.5 \times 10^{-4}$), whereas control mRNA genes contained no exonic nucleotides

corresponding to TEs (Fig. 7e). Unlike mice, where both sense and antisense TEs are embedded in uppl mRNAs, an antisense bias of TEs was detected in chickens (sense median = 0.0%, antisense median = 2.1%). LINE piRNAs produced in chicken adult testes manifest a 1U bias but not a 10A bias (Fig. 7f). These piRNAs, uniquely mapping to uppl mRNAs, can also target TEs in trans, post-transcriptionally generating secondary piRNAs (Fig. 7g). Although neither uppl genes nor TE families (Fig. 7a, e) are conserved between mice and chickens, which may accommodate rapidly changing populations of TEs, our data show that the use of mRNAs embedded with TEs to produce antisense TE piRNAs that cleave TEs post-transcriptionally is a common strategy in amniotes.

## Discussion

Here we systematically characterized 3′UTR piRNAs (which should be called genic piRNAs, as ORF regions also produce piRNAs). By defining the transcription factors associated with piRNA biogenesis and characterizing mutants with transcriptional and post-transcriptional processing defects, we demonstrate that full-length mRNAs are the precursors of 3′UTR piRNAs. These mRNAs undergo pioneer rounds of translation that are followed by the production of piRNAs. This coupling is mediated by post-termination 80S ribosomes on 3′UTRs that guide endonucleolytic cleavages and generate the 5′-ends of piRNAs. Together with our previous studies on piRNA biogenesis from lncRNA precursors[45], we find that ribosomes guide piRNA biogenesis downstream of ORFs regardless of ORF length. Similar to other co-translational mRNA quality-control pathways, piRNA processing from mRNAs fine-tunes the protein products from these mRNAs. This co-translational processing of piRNAs is found in both mice and chickens. Therefore, we reveal a general and conserved mechanism by which post-termination ribosomes guide piRNA 5′-end formation from non-protein-coding regions of RNAs in amniotes.

TE silencing in mice is thought to be carried out principally by prenatal piRNAs[124,125]. However, TE surveillance is still required throughout spermatogenesis. The murine PIWI protein MILI is not detected after prenatal piRNA expression ends and before pachytene piRNA expression begins[126], resulting in a loss of production of prenatal piRNAs and potentially leaving TEs unrestrained after birth and before pachynema. Indeed, we observed a very significant increase in the expression load of SINEs from 10.5 dpp to 20.5 dpp. The expression of 3′UTR piRNAs throughout spermatogenesis (Supplementary Fig. 1a) may be critical for germline genome protection against activated TEs when TE piRNAs produced from other sources are low. Since TE piRNAs can also cleave other mRNAs that harbor TEs, as previously reported[127], the TE piRNAs produced from uppl mRNAs may lead to a cascade of events to regulate mRNA stability. Given the rapid changes in TE families in each species over evolutionary time, the shared mechanisms for producing TE piRNAs from distinct gene subsets in divergent lineages suggest the biological significance of this strategy.

Furthermore, we discovered that fine-tuning of essential protein levels is a function of 3′UTR piRNA processing. One target of co-translational piRNA processing in mice spermatocytes is Ago2 mRNAs. Consistent with the idea that piRNA processing attenuates AGO2 function, defective piRNA processing in Mov10l1 mutants leads to increased AGO2 protein levels and decreased protein synthesis of AGO2-miRNA targets. Moreover, most AGO2-miRNA targets in spermatocytes harbor essential functions, allowing the initial impact on AGO2 protein levels to be further amplified. Although it is technically challenging, if even possible, to single out the impact of each uppl protein change on

spermatogenesis, the combination of the dysregulation of these genes is likely to be a non-negligible contributor to the Mov10l1 mutant phenotype. While the set of protein-coding genes that produce piRNAs is not evolutionarily conserved between mice and chickens, the functions of uppl mRNA genes are in general critical for spermatogenesis and viability in mice and chickens. The high enrichment of chicken uppl mRNA genes from the Z chromosome might reflect the requirements for dosage compensation from the two Z chromosomes in males, given that in chickens females are ZW with only one Z chromosome and the piRNA pathway is not active in the chicken ovary[128]. Taken together, although fine-tuning protein is unlikely to represent the primary selective force that marks a subset of mRNAs as piRNA precursors, we expect the better fitness gained by the rapid turnover of these essential mRNAs to further reinforce their piRNA precursor identity.

The ribosome mitigation downstream of ORFs regardless of ORF length suggests compromised ribosome recycling. During conventional termination, ABCE1 binds to post-termination ribosomes and splits the large and small subunits[129,130]. After 60S dissociation, a complex including eIF2D/Tma64 (the homolog of eIF2D in yeast), MCST1/Tma20, and DENR/Tma22 is required for 40S ribosome recycling in yeast. In general, translation termination is a highly controlled process, with the binding of 80S ribosomes to 3′UTRs only being demonstrated in either terminally differentiated erythrocytes or in the context of mutated translational machinery[79–84,131–134]. In our study, the regulation of ribosome recycling occurs during meiosis prophase in cells that are not terminally differentiated, as spermatocytes will undergo two more rounds of cell division, weeks of further differentiation, and continued protein synthesis. This ribosome recycling regulation is specific to uppl mRNAs and may be due to the localization of uppl mRNAs in proximity to the mitochondria where piRNAs are processed. The process is enabled, at least partially, by the reprogramming of RNA metabolism at pachynema, including the inhibition of ribosome recycling and the NMD pathway, allowing for robust piRNA production in a short but critical time window during spermatogenesis. Our study indicates that reprogramming of ribosome recycling can occur locally and stage-specifically to enable biologically significant processes.

Our systematic characterization of 3′UTR piRNAs provides insights into the fundamental question: why are some transcripts processed to piRNAs while others are not? piRNA precursors could be either marked epigenetically during transcription or are recognized post-transcriptionally in the cytosol. Unlike Drosophila germline piRNA loci marked with a chromatin-bound protein Rhino[30], currently no epigenetic factors have been identified to specially bind to piRNA loci in mammals. Although splicing signature is unlikely to be the hallmark for piRNA precursors in mammals as proposed for Drosophila piRNA biogenesis[59], both lncRNA piRNA precursors and mRNA piRNA precursors exhibit longer first exons, suggesting that unique exon–intron structure could be one unique feature. Furthermore, given the translation of short upstream ORFs is insufficient to mark a transcript for piRNA biogenesis and the uppl mRNAs do not exhibit faulty translation on their main ORFs, the translation intermediates with post-termination ribosomes on a long 3′UTR is a prime candidate for further testing. Last but not least, if the TE-rich prenatal piRNAs target and initiate the processing of the 3′UTR piRNA precursors, the embedding of TE elements could also serve as a determining feature of a transcript for piRNA processing. Thus, the study of 3′UTR piRNAs allows a more comprehensive investigation into the unique features defining piRNA precursors.

The linked but distinct biogenic requirements before and after the stop codon of uppl main ORFs suggest that post-

transcriptional processing of piRNAs can be further divided into two phases: substrate recognition and efficient processing. Since uppl ORFs are still processed into piRNAs distinct from other cellular mRNAs, this suggests that uppl mRNAs are recognized as substrates for piRNA processing as entire transcripts before endonucleolytic cleavages occur. Otherwise, if the substrate recognition couples with piRNA processing, as soon as the initial cleavage starts on 3′UTRs, the ORF portion of the transcripts would display no unique feature compared to other translating mRNAs. Our finding that the ORF portion of transcripts still enter piRNA processing but in an inefficient manner suggests that either post-termination ribosomes streamline piRNA production and/or the translating ribosomes inhibit piRNA processing machinery from accessing the RNAs. The requirement of TDRD5 for 3′UTR piRNA biogenesis suggests that TDRD5 may function by coordinating post-termination ribosomes with piRNA processing machinery. TDRD5 detected in amniotes (which harbor a homolog with >50% protein sequence identity with the mouse TDRD5) but not in fish (<10% of potential homologs)[135] may have co-evolved with appearance of ribosome-guided piRNA biogenesis. Furthermore, although 3′UTR piRNAs have been characterized in fruit flies and the *cis* RNA elements sufficient to promote piRNA biogenesis in somatic cells have been identified[136,137], their biogenesis remains when upstream translation is inhibited[137], and the region downstream of the *cis* RNA element produce fewer piRNAs than the element region itself[136]. Thus, consistent with the recent appearance of TDRD5, ribosome-guided piRNA biogenesis is unlikely to be conserved in invertebrates and further studies are required to trace its evolutionary origins.

In summary, we reveal a conserved and general piRNA biogenesis mechanism that recognizes translating RNAs regardless of whether they harbor long ORFs or not. The assembly of 80S ribosomes on non-coding regions of RNA is not restricted by the length of the upstream ORFs and is temporally staggered with translation-dependent RNA quality-control pathways, suggesting compromised ribosome recycling. The coupling of piRNA biogenesis with translation fine-tunes the abundance of proteins that are critical for spermatogenesis in both mice and chickens.

## Methods

**Animals**. Mice were maintained and used according to guidelines for animal care of the NIH and the University Committee on Animal Resources at the University of Rochester. Mice of the following strains C57BL/6J (Jackson Labs, Bar Harbor, ME, USA; stock number 664); $Rpl22^{tm1.1Psam}$ on a C57BL/6J background (Jackson Labs; stock number 011029)[138]; $Mov10l1^{tm1.1Jw}$ on a mixed 129×1/SvJ × C57BL/6J background[11]; and $Tg(Neurog3-cre)$C1Able/J on a B6.FVB(Cg) background (Jackson Labs; stock number 006333)[139] were genotyped as described. Comparisons of compound mutants and controls involving *Mov10l1* CKO mutation were performed using siblings from individual litters. White Leghorn testes of the Cornell Special C strain from 1-year-old roosters were used according to guidelines for animal care of the NIH and the University Committee on Animal Resources at the University of Rochester.

**Small RNA sequencing library construction**. Small RNA libraries were constructed and sequenced, as previously described[45], using oxidation to enrich for piRNAs by virtue of their 2′-*O*-methyl-modified 3′ termini. The oxidation procedure selects against in vitro-digested products with a 3′ phosphate. A 25-mer RNA with 2′-*O*-methyl-modified 3′ termini (Supplementary Table 1, Spike-in RNA) was used as a spike-in control.

**Western blotting**. For protein detection, testis lysates were resolved by electrophoresis on 10% sodium dodecyl sulfate (SDS)-polyacrylamide gels. The proteins were transferred to a 0.45 μm polyvinylidene difluoride membrane (EMD Millipore, Billerica, MA, USA), and the blot was probed with anti-AGO2 mouse monoclonal antibody (Wako Pure Chemical Corporation, 018-22021, 1:500), anti-TUBULIN rabbit antibody (Bimake, Houston, TX, USA, A5105, 1:1000), and then detected with sheep anti-mouse immunoglobulin G–horseradish peroxidase (IgG-HRP; GE Healthcare, Little Chalfont, UK; NA931V, 1:5000), and donkey anti-

rabbit IgG-HRP (GE Healthcare, NA934V, 1:5000). Western blotting images were taken using Azure c300 imaging system, with the cSeries capture software (v1.6).

**Polysome gradient**. Fresh testes were lysed in 1 ml lysis buffer (10 mM Tris-HCl, pH 7.5, 5 mM $MgCl_2$, 100 mM KCl, 1% Triton X-100, 2 mM dithiothreitol (DTT), 100 μg/ml cycloheximide, and 1× protease-inhibitor cocktail) as previously described[45]. Five $A_{260}$ absorbance units were loaded on a 10–50% (w/v) linear sucrose gradient prepared in buffer (20 mM HEPES-KOH, pH 7.4, 5 mM $MgCl_2$, 100 mM KCl, 2 mM DTT, 100 μg/ml cycloheximide) and centrifuged in a SW-40ti rotor at 154,348 × *g* for 2 h 40 min at 4 °C. Samples were collected from the top of the gradient using a gradient Fractionation system (Brandel, Boca Raton, FL, USA; BR-188) while monitoring absorbance at 254 nm. Synthetic spike-in RNAs were added to each collected fraction before RNA purification. RNA was purified by Trizol (Ambion, Waltham, MA, USA) using the Direct-zol™ RNA MiniPrep plus Kit (Zymo Research, Irvine, CA, USA).

**Ribosome profiling**. Ribo-seq was performed as previously described[45]. Cleared testis lysates were incubated with 60 units of RNase T1 (Fermentas, Waltham, MA, USA) and 100 ng of RNase A (Ambion) per $A_{260}$ unit for 30 min at room temperature. Samples were loaded on sucrose gradients, and after centrifugation, the fractions corresponding to 80S monosomes were recovered for library construction.

The 5′P and 5′OH Ribo-seq library was prepared as follows (Supplementary Fig. 4a). (i) Size-selected and rRNA-depleted RPFs were ligated to a 5′ adapter. The ligation products were resolved on a 15% denaturing gel, and ligated products and unligated RNAs were purified separately. (ii) The 3′-ends of recovered RNAs were dephosphorylated and ligated to 3′ adapters (Supplementary Table 1, 3′ adapter). (iii) The unligated RNAs proceed to the conventional Ribo-seq library construction. (iv) The ligated RNAs directly proceed to the reverse transcription steps for library construction.

**RNA sequencing**. Strand-specific RNA-seq libraries were constructed following the TruSeq RNA sample preparation protocol, as previously described[45]. rRNAs were depleted from total RNAs with complementary DNA oligomers (IDT) and RNase H (Invitrogen, Waltham, MA, USA)[140,141]. RNA-seq data were generated on HiSeq 2000 instrument (Illumina, San Diego, CA, USA).

**Degradome-seq library construction**. Degradome-seq library construction was performed as described previously[45]. The RNAs were first oxidized at room temperature for 30 min with sodium periodate (Sigma, St. Louis, MO, USA) to block the 3′-ends from ligation and were then size-selected to isolate RNA ≥200 nts (DNA Clean & Concentrator™-5, Zymo Research). 5′ Adapters were attached using T4 RNA ligase (Ambion) at 20 °C for 3 h. The ligated products were subjected to rRNA depletion with complementary DNA oligomers (IDT) and RNase H (Invitrogen)[140,141]. The rRNA-depleted ligation products were reverse transcribed using a degenerate primer (Supplementary Table 1, Degenerate primer). cDNA was amplified by PCR using KAPA HIFI Hotstart polymerase (Kapa Biosystems, Wilmington, MA, USA), and 250–350 nts double-stranded DNA was isolated on 8% native PAGE gels.

**Histology and immunostaining**. For histologic analysis, testes were fixed in Bouin's solution overnight, embedded in paraffin, and sectioned at 4 μm. Following standard protocols, sections were deparaffinized, rehydrated, and then stained with hematoxylin and eosin.

PELOTA/DOM34 immunostaining and mouse monoclonal anti-HA antibody (ascites fluid, 1:2000 dilution; Covance, Princeton, NJ, USA; MMS-101P) were performed on squashed spermatocytes and spermatids as previously described[142]. Seminiferous tubules were fixed in 2% paraformaldehyde containing 0.1% Triton X-100 for 10 min at room temperature, placed on a slide coated with 1 mg/ml poly-L-lysine (Sigma) with a small drop of fixative, gently minced with tweezers, and squashed. The coverslip was removed after freezing in liquid nitrogen. The slides were later rinsed three times for 5 min in phosphate-buffered saline (PBS) and incubated for 12 h at 4 °C with rabbit anti-PELOTA antibody (1:50 dilution; Thermo, PA5-31697). Secondary antibodies conjugated with Alexa Fluor 488 (Molecular Probes, Eugene, OR, USA) were used at a dilution of 1:500. Histology and immunostaining images were taken using Leica DM4000 B LED microscope system with the Leica software: Leica Application Suite X v1.1.0.12420.

**IP from mouse testes**. For CBP80 and eIF4E IPs, anti-CBP80 rabbit antibody (Bethyl Laboratories, A301-794A) or anti-eIF4E rabbit antibody (Bethyl Laboratories, A301-153A) from mouse testis lysates were performed as described[143], except Protein A magnetic Beads (161-4013, Bio-Rad, Hercules, CA, USA) were used.

**Intra-testicular injection**. The mice were anesthetized with ketamine/xylazine mixture (ketamine 100 mg/kg; xylazine 25 mg/kg) via intraperitoneal injection. After complete anesthesia, testes were exteriorized with a longitudinal incision around 1 cm at the center of abdomen. The tunica albuginea was penetrated using

a sharp 26 G needle (BD, Franklin Lakes, NJ, USA, 30511) 1 mm from the vascular pedicle, and the needle was withdrawn to generate a path for introducing a blunt end Hamilton needle (Hamilton, Reno, NV, USA, 7786-02). PBS containing 0.02% Fast Green FCF (Thermo Fisher Scientific, Waltham, MA, USA) with harringtonine (LKT labs, 0.5 µg/µl in a total volume of 10 µl) or with Okadcid Acid (LX Laboratories, Woburn, MA, USA, O-2220, 16 nM per testis in a total volume of 10 µl) was slowly injected using a Hamilton microsyringe (1705RN) into one testis, and a vehicle control without the drug was injected into the other testis of the same animal. The needle was held in place for 30 s before removal to prevent backflow of the solution. Successful completion of injection was indicated by testis filled with green solution. The testes were returned to the abdominal cavity after injection. The incisions were sutured. At the end point, the mice were euthanized by cervical dislocation, and the testes were collected.

**General bioinformatics analyses**. Analyses were performed using piPipes v1.4[144]. All data from the small RNA sequencing, Ribo-seq, 40S footprinting, 80S footprinting, RNA-seq, degradome-seq, CLIP sequencing, and ChIP-seq were analyzed using the latest mouse genome release mm10 (GCA_000001635.7) and chicken genome release galGal6 (GCA_000002315.5). Generally, one mismatch is allowed for genome mapping. For mouse transcriptome annotation, 30 uppl mRNAs defined in our previous studies with mm9[8] were converted to mm10 coordinates with using liftOver[145] with minor manual correction (Supplementary Data 1). We selected 43 control mRNAs (Supplementary Data 1) from our recently reconstructed mouse testis transcriptome[60] by their similar expression dynamics as uppl mRNAs from 10.5 dpp to 20.5 dpp and by their lack of any piRNA production. For genes with alternative transcripts, the most abundant transcript involving that gene was selected. We reassembled mRNAs using RNA-seq data from rooster testes and have defined 3′UTR piRNA precursor mRNAs as described in the text. Statistics pertaining to the high-throughput sequencing libraries constructed for this study are provided in Supplementary Data 1.

For small RNA-seq, libraries were normalized to the sum of total miRNA reads; spike-in RNA was used to normalize the libraries from each fraction of polysome profiling. Uniquely mapping reads >23 nts were selected for further piRNA analysis. We analyzed previously published small RNA libraries from adult wild-type mouse testis (GSM1096604)[8], mouse Tdrd5 CKO, control testis (SRP093845)[78], A-Myb mutant, control testis at 14.5 and 17.5 dpp (GSM1096588, GSM1096589, GSM1096590, GSM1096591, GSM1096605, GSM1096606, GSM1096607, GSM1096608), RNase treated (GSM4160780) and untreated adult testis (GSM4160781), Mov10l1 CKO mutants (GSM4160774, GSM4160775, GSM4160776, GSM4160777, GSM4160778, and GSM4160779), and littermate controls (GSM4160768, GSM4160769, GSM4160770, GSM4160771, GSM4160772, and GSM4160773)[45]. We report piRNA abundance either as parts per million reads mapped to the genome (ppm), or as reads per kilobase pair per million reads mapped to the genome (rpkm) using a pseudo count of 0.001. To ensure precision of the mapping, a piRNA is counted only when the 5′-end of the piRNA maps to the ORF or 3′UTR of a transcript.

For RNA-seq reads, the expression per transcript was normalized to the top quartile of expressed transcripts per library calculated by Cufflinks v2.2.1[146], and the transcripts per million value was quantified using the Salmon v0.8.2 algorithm[147]. We analyzed previously published RNA-seq libraries from mouse A-Myb mutant, control testis at 14.5 and 17.5 dpp (GSM1088426, GSM1088427, GSM1088428, GSM1088429), wild-type mouse testis at 10.5 dpp, 12.5 dpp, 14.5 dpp, 17.5 dpp, 20.5 dpp, and adult stage (GSM1088421, GSM1088422, GSM1088423, GSM1088424, GSM1088425, and GSM1088420)[8], Mov10l1 CKO mutants (GSM4160761, GSM4160762, and GSM4160753), and littermate controls (GSM4160758, GSM4160759, and GSM4160760)[45].

Ribo-seq analysis followed the modified small RNA pipeline including the junction mapping reads as previously described[45]. Uniquely mapping reads between 26 and 32 nts were selected for further analysis except for the analysis on the 40S footprints where 18–80 nts were selected. RPFs and 80S footprints from different developmental stages were normalized to the sum of reads mapping to mRNA protein-coding regions, assuming that mRNA translation was largely unchanged during spermatogenesis. Libraries from harringtonine treatment were further normalized to the sum of reads mapping to mitochondrial coding sequences (CDSs) as previously described[148]. 40S footprints were normalized to the sum of reads mapping to mRNA 5′UTRs as it has been shown that the 40S binds 5′ UTRs in a cap-tethered fashion, thus 40S ribosomes do not accumulate upon harringtonine treatment[66]. We analyzed published Ribo-seq libraries from Mov10l1 CKO mutants (GSM4160728, GSM4160729, and GSM4160730) and litter mate controls (GSM4160725, GSM4160726, and GSM4160727), anti-HA IP RPF (GSM4160731, GSM4160732, and GSM4160733) and input (GSM4160734, GSM4160735, and GSM4160736) from RiboTag mice, wild-type adult testis injected with (GSM4160744 and GSM4160745) and without harringtonine (GSM4160746), in vitro sheared RNA fragments (GSM4160749), monosome (GSM4160752), polysome (GSM4160753), adult testis (GSM4160756 and GSM4160757)[45], and adult rooster wild-type testis (GSM2454692)[128].

ChIP-seq reads were analyzed as previously described[8]. Multiple mapping reads were apportioned randomly to each location (-k 1 switch) and one mismatch was allowed (-v 1). ChIP peaks were identified using MACS2 v2.1.1.20160309[149] using default arguments and input as control. We analyzed published A-MYB ChIP-seq

(GSM1087281) and input DNA libraries (GSM1087284) from mouse wild-type adult testis and A-MYB ChIP-seq (GSM1087285), H3K4me3 ChIP-seq (GSM1087286), and input DNA libraries (GSM1087287) from rooster wild-type adult testes[8].

Degradome reads and CLIP sequencing reads were aligned to the genome using TopHat 2.0.12[150]. Reads were mapped uniquely using the "-g 1" flag. Uniquely mapping reads were selected for further analysis. Libraries were normalized to the sum of reads mapping to mRNA protein-coding regions, assuming that mRNA cleavage was largely unchanged during spermatogenesis. We analyzed a published anti-HA IP degradome library from adult wild-type testis (GSM4160721)[128] and MOV10L1 CLIP library mouse wild-type testis (PRJNA230507)[13].

Statistical analyses were performed in R v3.5.0[151]. The significance of the differences was calculated by Wilcoxon rank-sum test except as indicated in the text. The significance of correlation was performed using partial correlation analysis in addition to simple correlations[72].

**Chicken transcriptome assembly and annotation**. We used StringTie v1.3.3b[152] with parameter "-m 50 –p 8 –G RefSeq –rf –l prefix" to assemble transcripts from RNA-seq reads from chicken testes. After assembling each sample, StringTie's merge function was used to merge each assembly. We required the transcript abundance to be at least 8 fpkm and filtered 406,332 assembled transcripts to 39,940 transcripts. We further added transcripts that are not covered in our own assembly but reported in the latest RefSeq (downloaded on Feb 19, 2020), resulting in 81,079 total transcripts. Finally, using TransDecoder v5.5.0[153] with the BlastP (v2.10.0 +) and Hmmer (v3.3) search, we identified 44,856 mRNA transcripts and considered the remaining 36,223 transcripts as lncRNAs.

**5′-end overlap analysis and phasing analysis**. Reads were mapped to the transcriptome, and their relative positions on transcripts were reported. For alternative transcription with overlapping annotations, we chose the longest transcript. We calculated the distance spectrum of 5′-ends of Set A (RPFs or degradome reads) that overlapped with Set B (piRNAs or simulated sequences) as follows: for each read b in Set B, we identified all the reads in Set A whose 5′-ends overlapped within the 50-nt region upstream of b including the 5′-end of b and 50 nts downstream of b (200-nt window of b reads). We assigned the b spectrum as the fractions of 5′-ends of a reads distributed across the 100-nt window of b reads. The height of the b spectrum at each nucleotide position in the 100-nt window of b reads was based on the number of a reads whose 5′-ends overlapped at this position divided by the total number of a reads whose 5′-ends overlapped with the 100-nt window of b reads. The sum of all b spectra was then divided by the total number of reads in Set B. We defined this average fraction of 5′-ends of a reads that overlapped with the 100-nt window of b reads as the distance spectrum of 5′-ends of Set A that overlap Set B. The Z score for overlap at the 5′-end position was calculated using the spectral value at positions −50−−1 and 2–50 as background.

**Nucleotide periodicity**. Nucleotide periodicity was computed as previously described[45]. We first aligned the RPFs to each other using 5′-end overlap analysis and reported the distance spectrum. An annotated ORF was not a prerequisite for this analysis as the distance spectrum of RPFs from mRNAs already showed a 3-nt periodicity pattern. We then transformed the distance spectrum using the "periodogram" function from the GeneCycle v1.1.4 package[154] with the "clone" method. The relative spectral density was calculated by normalizing to the value at the first position.

**Generating simulated sequences as negative controls**. We generated a random pool of 28-mer sequences using a sliding window of 1 nucleotide from 5′ to 3′ of the piRNA precursors. We then sampled from this 28-mer pool to match the first nucleotide composition of the real reads. These simulated sequences from piRNA precursors were used as random controls for piRNAs (source code available upon request).

**Mass spectrometry sample preparation and analysis**. After testes lysis, protein concentration was determined by BCA (Thermo Scientific). Samples were then diluted to 1 mg/ml in 5% SDS, 50 mM triethylammonium bicarbonate (TEAB). In all, 25 µg of protein from each sample was reduced with DTT to 2 mM, followed by incubation at 55 °C for 60 min. Iodoacetamide was added to 10 mM and incubated in the dark at room temperature for 30 min to alkylate the proteins. Phosphoric acid was added to 1.2%, followed by six volumes of 90% methanol and 100 mM TEAB. The resulting solution was added to S-Trap micros (Protifi) and centrifuged at 4000 × g for 1 min. The S-Traps containing trapped proteins were washed twice by centrifuging through 90% methanol and 100 mM TEAB. One microgram of trypsin was brought up in 20 µl of 100 mM TEAB and added to the S-Trap, followed by an additional 20 µl of TEAB to ensure the sample did not dry out. The cap to the S-Trap was loosely screwed on but not tightened to ensure the solution was not pushed out of the S-Trap during digestion. Samples were placed in a humidity chamber at 37 °C overnight. The next morning, the S-Trap was centrifuged at 4000 × g for 1 min to collect the digested peptides. Sequential additions of 0.1% trifluoroacetic acid (TFA) in acetonitrile and 0.1% TFA in 50% acetonitrile

were added to the S-trap, centrifuged, and pooled. Samples were frozen and dried down in a Speed Vac (Labconco), then re-suspended in 0.1% TFA prior to analysis.

Peptides were loaded onto a 100 µm × 30 cm C18 nano-column packed with 1.8 µm beads (Sepax), using an Easy nLC-1200 HPLC (Thermo Fisher) connected to a Orbitrap Fusion Lumos mass spectrometer (Thermo Fisher). Solvent A was 0.1% formic acid in water, and solvent B was 0.1% formic acid in 80% acetonitrile. Ions were delivered to the mass spectrometer using a Nanospray Flex source operating at 2 kV. Peptides were eluted off the column using a multi-step gradient, which started at 3% B and held for 2 min, quickly ramped to 10% B over 7 min, increased to 38% B over 152 min, then ramped up to 90% B in 6 min and held there for 4 min to wash the column before returning to starting conditions in 2 min. The column was re-equilibrated for 7 min for a total run time of 180 min. The flow rate was 300 nl/min. The Fusion Lumos was operated in data-dependent mode, performing a full scan followed by as many MS2 scans as possible in 3 s. The full scan was done over a range of 375–1400 $m/z$, with a resolution of 120,000 at $m/z$ 200, an AGC target of 4e5, and a maximum injection time of 50 ms. Peptides with a charge state between 2 and 5 were selected for fragmentation. Precursor ions were fragmented by collision-induced dissociation using a collision energy of 30 and an isolation width of 1.1 $m/z$. MS2 scans were collected in the ion trap with the scan rate set to rapid, a maximum injection time of 35 ms, and an AGC setting of 1e4. Dynamic exclusion was set to 45 s.

Raw data were searched using SEQUEST within the Proteome Discoverer software platform, v2.2 (Thermo Fisher) employing the SwissProt mouse database, along with a custom fasta database that included both test and control proteins. Trypsin was selected as the enzyme allowing up to 2 missed cleavages, with an MS1 mass tolerance of 10 ppm, and an MS2 mass tolerance of 0.6 Da. Carbamidomethyl on cysteine was selected as a fixed modification. Oxidation of methionine was set as a variable modification. A percolator was used as the false discovery rate calculator, filtering out peptides with a $q$ value > 0.01. Label-free quantitation was performed using the Minora Feature Detector node with a minimum trace length of 5. The Precursor Ions Quantifier node was then used to calculate protein abundance ratios using only unique and razor peptides. The summed abundance-based method was employed, which sums the peak areas for all the peptides for a given protein to determine protein ratios.

**Codon Adaptation Index (CAI) analysis**. CAI was calculated for coding region of uppl mRNAs and control mRNAs using two different software packages: DAMBE v7.2.1[155] and CAIcal v1.4[156]. Both DAMBE software package by Xia et al. and CAIcal from Puigbo's group use alternative implementation[70] of Sharp's formula[69] to calculate CAI. CAI calculates normalized CAI as quotient between the CAI of the query sequence and expected CAI (eCAI) of 1000 randomly generated sequences with G + C and amino acid content similar to that of the query sequence[71]. Random sequences were generated using the Markov method, and eCAI was estimated at 99% level of confidence and 99% coverage. CDSs of housekeeping genes were used as the reference set for all the calculations.

**miRNA target search**. miRNA target search was performed by miRanda v3.3[157] using the -strict parameter.

**Reporting summary**. Further information on research design is available in the Nature Research Reporting Summary linked to this article.

## Data availability

The data supporting the findings of this study are available from the corresponding author upon reasonable request. Next-generation sequencing data used in this study have been deposited at the NCBI Gene Expression Omnibus under the accession number GSE155350. Mass spectrometry data have been uploaded to the ProteomeXchange Consortium via the PRIDE database under accession number PXD027489. Source data are provided with this paper.

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

## Acknowledgements
We thank L. Maquat, T. Kurosaki, D. Ermolenko, J. Lin, X. Zhuo, and members of the Li laboratory for discussions; G. Riddihough and J. Lozada for help with editing the manuscript; and J. P. Wang, UR pathology core, URMC Mass Spectrometry Resource Facility, and UR Genomics Research Center for help with the experiments. We thank N. Chen for creating the cartoon pictures. This work was supported by National Institutes of Health grant R35GM128782 and Agriculture and Food Research Initiative Competitive Grant no. 2018-67015-27615 from the USDA National Institute of Food and Agriculture to X.Z.L.

## Author contributions
Y.H.S. and J. Zheng analyzed the data with input from C.Z., E.P.R., and X.Z.L.; R.H.W, K.D., J. Zhu, L.H.X., and A.A.P. performed the experiments with input from E.P.R. and X.Z.L. X.Z.L. designed the study and drafted the manuscript, and all authors contributed to the preparation of the manuscript.

## Competing interests
The authors declare no competing interests.
