## [Peer Review File · Nature Communications]

Coupled protein synthesis and ribosome-guided piRNA processing on mRNAsREVIEWERS' COMMENTS

Reviewer #1 (Remarks to the Author):

This revised manuscript is quite a remarkable work, demonstrating the ribosome-guided piRNA biogenesis from 3' UTR of mRNAs. Building upon the authors' prior work, this study brings up several novel points. Compared to the original manuscript, the authors did an excellent job in addressing reviewers' concerns. The additional experiments strengthened the conclusion. I think this revised manuscript is now suitable for Nat Communications.

Reviewer #2 (Remarks to the Author):

This reviewer appreciates the author's effort in revising the manuscript to address the reviewers' comments. The changes in response to my comments are satisfactory. I support the publication of the revised manuscript with the following a minor addition:

We are still left with essentially no mechanisms for how some transcripts are processed to piRNAs while others are not. What could act as a bridging component between ribosomes on 3' UTRs and the piRNA producing machinery? Some years ago, Homolka et al. (Cell Reports 12, 418-428, 2015) and Ishizu et al. (Cell Reports 12, 429-440, 2015) identified cis-acting RNA elements on 3' UTRs that direct genic piRNA production in *Drosophila*. These studies can be discussed.

Response to Reviewers' Comments

NCOMMS-21-26347

" Coupled protein synthesis and ribosome-guided piRNA processing on mRNAs"

We thank the editor for evaluating our study and the Reviewers for providing constructive comments that have helped improve our manuscript. In this work, we investigated whether ribosomes guide piRNA biogenesis from piRNA precursors with long open reading frames (ORFs). We systematically characterized the biogenesis, regulation, and function of 3'UTR piRNAs. We demonstrated that their precursors are full-length mRNAs and that ribosome-guided piRNA biogenesis occurs after translation of the main ORFs on these mRNAs. The 3'UTR piRNAs are transcriptionally regulated by the A-MYB transcriptional factor, and their post-transcriptional piRNA processing guided by ribosomes is enabled by the sequestration of other translation-dependent decay pathways at the pachytene stage. The biogenesis of 3'UTR piRNAs fine-tunes protein amounts, and one outcome of this function is crosstalk with the miRNA pathway. We are pleased to see that both the Reviewers recognize the importance of our findings and the quality of our research. In this revision, we have addressed the reviewers' comments, and included the access numbers for our mass spectrometry data. We hope the editor and reviewers agree that the revised MS is now worthy of publication in *Nature Communications*.

Reviewer #1 (Remarks to the Author):

This revised manuscript is quite a remarkable work, demonstrating the ribosome-guided piRNA biogenesis from 3' UTR of mRNAs. Building upon the authors' prior work, this study brings up several novel points. Compared to the original manuscript, the authors did an excellent job in addressing reviewers' concerns. The additional experiments strengthened the conclusion. I think this revised manuscript is now suitable for Nat Communications.

Response: We appreciate the reviewer's comments.

Reviewer #2 (Remarks to the Author):

This reviewer appreciates the author's effort in revising the manuscript to address the reviewers' comments. The changes in response to my comments are satisfactory. I support the publication of the revised manuscript with the following a minor addition:

We are still left with essentially no mechanisms for how some transcripts are processed to piRNAs while others are not. What could act as a bridging component between ribosomes on 3' UTRs and the piRNA producing machinery? Some years ago, Homolka et al. (Cell Reports 12, 418-428, 2015) and Ishizu et al. (Cell Reports 12,

429-440, 2015) identified cis-acting RNA elements on 3' UTRs that direct genic piRNA production in *Drosophila*. These studies can be discussed.

Response: We appreciate the reviewer's comment and have included the references in the discussion as the following:

“Furthermore, although 3'UTR piRNAs have been characterized in fruit flies and the cis RNA elements sufficient to promote piRNA biogenesis in somatic cells have been identified^{1,2}, their biogenesis remains when upstream translation is inhibited², and the region downstream of the cis RNA element produce fewer piRNAs than the element region itself¹.”

In summary, we believe our study presents a substantial mechanistic advance in the field as we uncover a conserved and general ribosome-guided piRNA biogenesis pathway regardless of their precursor types, which would require special regulation of ribosome recycling and inhibition of other translational dependent RNA quality control mechanisms. We discovered a new function of the piRNA pathway in fine-tuning the production of proteins essential for reproduction and discussed the unique features of piRNA precursors in mammals. While the complete understanding of these mechanisms requires extensive additional work, we believe our manuscript will stimulate new avenues of research aimed at answering the exciting questions surrounding ribosomes, piRNAs, and germ cells.

1. Homolka, D. et al. PIWI slicing and RNA elements in precursors instruct directional primary piRNA biogenesis. *Cell Rep* **12**, 418-428 (2015).
2. Ishizu, H. et al. Somatic Primary piRNA Biogenesis Driven by cis-Acting RNA Elements and trans-Acting Yb. *Cell Rep* **12**, 429-440 (2015).